# Towards a Theoretical Understanding of the 'Reversal Curse' via Training Dynamics

**Hanlin Zhu***
UC Berkeley
hanlinzhu@berkeley.edu

**Baihe Huang***
UC Berkeley
baihe_huang@berkeley.edu

**Shaolun Zhang**
UC Berkeley
shaolun_zhang@berkeley.edu

**Michael Jordan**
UC Berkeley
jordan@cs.berkeley.edu

**Jiantao Jiao**
UC Berkeley
jiantao@berkeley.edu

**Yuandong Tian**
Meta AI
yuandong@meta.com

**Stuart Russell**
UC Berkeley
russell@cs.berkeley.edu

## Abstract

Auto-regressive large language models (LLMs) show impressive capacities to solve many complex reasoning tasks while struggling with some simple logical reasoning tasks such as inverse search: when trained on "$A \rightarrow B$" (e.g., *Tom is the parent of John*), LLM fails to directly conclude "$B \leftarrow A$" (e.g., *John is the child of Tom*) during inference even if the two sentences are semantically identical, which is known as the "reversal curse". In this paper, we theoretically analyze the reversal curse via the training dynamics of (stochastic) gradient descent for two auto-regressive models: (1) a bilinear model that can be viewed as a simplification of a one-layer transformer; (2) one-layer transformers under certain assumptions. Our analysis reveals that for both models, the reversal curse is a consequence of the (effective) model weights *asymmetry*, i.e., the increase of weights from a token $A$ to token $B$ during training does not necessarily cause the increase of the weights from $B$ to $A$, which is caused by the training dynamics under certain choice of loss function and the optimization space of model parameters. Moreover, our analysis can be naturally applied to other logical reasoning tasks such as chain-of-thought (COT), which provides a new perspective different from previous work that focuses on expressivity. Finally, we conduct experiments to validate our theory on multi-layer transformers under different settings. Our code is available at https://github.com/marlo-z/reversal_curse_analysis/.

## 1 Introductions

Large language models (LLMs) have shown great performance in solving complex reasoning tasks that require multiple reasoning steps through in-context learning (ICL), such as zero-shot learning [1, 2], few-shot learning [3, 4, 5], or via further fine-tuning [6, 7, 8]. However, without the above inference-time techniques or model fine-tuning (probably combined with data manipulations), an auto-regressive LLM might struggle with simple logical reasoning tasks that require multiple reasoning steps learned during training separately [9], where the reversal curse [10] serves as a well-known example.

---

*Equal contributions.

38th Conference on Neural Information Processing Systems (NeurIPS 2024).

The reversal curse refers to the phenomenon that an auto-regressive LLM that learns "$A \to B$" (e.g., *Tom is the parent of John*) during training fails to generalize to the reverse direction "$B \leftarrow A$" (e.g., *John is the child of Tom*) even if the pair of relationship "$\to$" and "$\leftarrow$" are reverse to each other and the two sentences are semantically identical. Although some previous works propose different methods to mitigate the reversal curse, including reversing the training dataset [11, 12] and training on different objectives such as autoregressive blank infilling [13], these methods might negatively affect the model performance on other tasks since they either alter the dataset or the model architecture. Without dataset manipulation or changing the auto-regressive nature (causal structure) of the model, there are two other candidate solutions to mitigate the reversal curse.

First, one might constrain the model parameters to satisfy a higher-level regularity for specific relationships. For example, a reversal-type regularity can be viewed as a pair of relationships ($\to$, $\leftarrow$) and two sets $\mathcal{A}, \mathcal{B}$ such that a model trained on "$A \to B$" will also increase its probability of "$B \leftarrow A$" for all $A \in \mathcal{A}, B \in \mathcal{B}$, which induces a subspace of model parameters that satisfy this regularity. If one can train the model within this subspace, then training on "$A \to B$" can, by definition, help to learn "$B \leftarrow A$". However, for a general LLM, it is extremely challenging to find the subspace and manually hard-code the constraint during optimization even for one pair of relationships, not to mention there are numerous relationships. Since it is intractable to manually hard-code the constraints to the model parameter, one can alternatively expect the model to learn the higher-level regularity by training samples under unconstrained optimization. However, this is also hard, according to our analysis, through the popular cross-entropy (CE) loss that aims to maximize the next token prediction probability for the models studied in our paper.

Second, one can use a different loss function which is "symmetric", rather than the popular CE loss. However, the "symmetric" loss might drive the model to learn meaningless sentences. For example, when trained on the sentence "John is tall", a "symmetric" loss function might drive the model to learn "tall is John", which is not what we expect. To prevent the model from the above undesired behavior, in practice, CE loss is still widely-used.

Therefore, in this paper, we analyze the reversal curse via training dynamics of the widely-used unconstrained optimization for the CE loss. We summarize our main contributions as follows:

- We theoretically analyze the reversal curse where training or test sequences have the form "$A \to B$" or "$B \leftarrow A$" via training dynamics of (stochastic) gradient descent under two auto-regressive models: a bilinear model (Section 3) and one-layer transformers under certain assumptions similar to [14] (Section 4). The analysis of both models reveals that the widely-used unconstrained optimization for CE loss leads to model weights *asymmetry*, i.e., the increase of (effective) weights (after reparameterization) from the token $A$ to token $B$[1] during training does not necessarily cause the increase of the weights from $B$ to $A$, which further causes the reversal curse. Although the (effective) weights from $A$ to $B$ and from $B$ to $A$ might be related to some extent due to reparameterization, their correlation is weak and thus show asymmetry as empirically verified in Section 5.

- The techniques we used to analyze the reversal curse can be applied to other logical reasoning tasks. In particular, we use the above framework to analyze chain-of-thought (COT) [4], and we show that a model trained on "$A \to B$" and "$B \to C$" separately struggles to directly conclude "$A \rightsquigarrow C$" without COT even if it is logically true (Section 4.2). Different from the previous work [15] that theoretically studies COT through the expressivity of transformers, our work provides a new perspective through training dynamics.

- We also empirically validate our theoretical results on multi-layer transformers (Section 5).

The *asymmetry* of auto-regressive model weights caused by widely-used unconstrained optimization for CE loss indicates that auto-regressive LLMs might not automatically deduce certain types of conclusions using separate knowledge learned during training under current popular training paradigms: to make a model predicting token $B$ where the input token is $A$, the model might need to see $B$ following $A$ in the same sequence during the training set. This also highlights the importance of ICL, data augmentation, or planning for LLMs with the current popular causal transformer-based structures to solve complex reasoning tasks.

---

[1] The weights from $A$ to $B$ can be viewed as the logits of token $B$ when the input is $A$.

## 1.1 Related works

**LLM Reasoning.** The strong performance of LLMs on reasoning tasks [3, 7, 16, 2, 17, 4, 18, 19] has prompted many studies on the reasoning capabilities of LLMs. [20] argues that transformers perform implicit Bayesian inference in ICL. [21] shows that transformers implement a specific type of circuits called "induction heads" that are key to the ICL abilities of LLMs. [22] proves that causal structures are encoded in transformer layers during the training dynamics. [23] identifies a backward chaining mechanism of transformers in deductive reasoning. Apart from in-context reasoning, LLMs still demonstrate limitations in other types of reasoning tasks [24, 25, 26].

**Reversal Curse.** [10] identifies the phenomenon of reversal curse. This drawback of LLMs is also demonstrated in [27]. [9] studies a similar phenomenonin which LLMs face difficulty in manipulating already learned knowledge. Several paper studies eliminating the reversal curse by extending causal attention to bidirectional attention [13], training on reversed samples [12], permuting semantic units [11], or introducing reverse logic data [28]. Given all the empirical works, theoretical analysis of the reversal curse phenomenon remains scarce.

**Expressivity of LLMs.** There is a long line of works [29, 30, 31, 32, 33, 34, 35, 36, 37, 38, 39, 40, 41, 42, 43, 44, 21, 45] studying the behavior of LLMs through the expressivity of transformers. It has been shown that transformers can implement simple functions such as sparse linear functions, two-layer neural networks, and decision trees [42], gradient descent [37, 44, 46], automata [47], Turing machines [48], variational inference [49], and bandit algorithms [50]. Different from [15] that study COT via expressivity, we analyze reversal curse and COT via training dynamics.

**Training dynamics of LLMs.** There are rich literatures in the optimization of attention layers [51, 52, 53, 54, 55, 56, 57]. [58, 59] study the dynamics of a single linear attention layer in in-context linear regression. [60] proves convergence of one-layer transformers in random feature regime. [61] shows the convergence of gradient descent on one-layer transformers in in-context linear regression with orthogonal data. [14] studies the convergence of one-layer transformers in a class of next-token prediction tasks. [62] studies training dynamics of multi-layer transformers. [22] studies gradient descent on a class of two-layer transformers in in-context learning tasks with latent causal structures. Our paper studies the reversal curse via training dynamics under both bilinear settings and one-layer transformers. For one-layer transformers, we use the same framework as [14] without the need for certain technical assumptions such as long input sequences, different learning rates for different parameters (except for Appendix C.3), or weak correlations that are required for [14]. Besides, we focus on the generalization ability of models for logical reasoning tasks while [14] mainly focus on optimization, and we identify the asymmetry and intransitivity properties of model weights, which are the core reasons for the failure of LLM for certain types of logical reasoning. Moreover, our analysis of the bilinear model only requires the embedding to be almost orthonormal, while [14] essentially assumed the embedding vectors to be fixed and one-hot.

## 2 Preliminaries

**Basic notations.** For any integer $N > 0$, we use $[N]$ to denote the set $\{1, 2, \ldots, N\}$. Let $\mathbb{R}$, $\mathbb{N}$ denote the set of real numbers and natural numbers, respectively. For real variables $x_1, \ldots, x_n$, we use $\text{poly}(x_1, \ldots, x_n)$ to denote the polynomial of $x_1, \ldots, x_n$. We use $f(n) \lesssim g(n)$ if there exists a constant $C > 0$ s.t. $f(n) \leq Cg(n), \forall n$; we say $g(n) \gtrsim f(n)$ if $f(n) \lesssim g(n)$.

We use $\boldsymbol{e}_i$ to denote one-hot vectors where only the $i$-th entry of $\boldsymbol{e}_i$ equals one and all other entries are zero. We use $\boldsymbol{1}$ to denote all-one vectors, $\boldsymbol{0}$ to denote zero vectors or zero matrices, and $I$ to denote the identity matrix. We will also add subscripts when we want to explicitly show the dimension, such as $\boldsymbol{0}_d, I_d$ for $d$-dimensional zero vector and $d \times d$ identity matrix. We use $\otimes$ to denote tensor product of vectors or matrices and use $\boldsymbol{x}^{\otimes 2}$ and $A^{\otimes 2}$ to denote $\boldsymbol{x} \otimes \boldsymbol{x}$ and $A \otimes A$ for vector $\boldsymbol{x}$ and matrix $A$.

We use $\mathcal{N}(\boldsymbol{\mu}, \Sigma)$ (or adding subscripts such as $\mathcal{N}_d(\cdot, \cdot)$ if we want to show dimensions explicitly) to denote the (multi-variate) Gaussian distribution with mean $\boldsymbol{\mu}$ and covariance $\Sigma$. Also, we use $\Delta(\mathcal{X})$ to denote the set of distributions over a set $\mathcal{X}$ and use $\mathbb{E}[\cdot]$ to denote expectation. For any dataset $\mathcal{D} = \{x_1, x_2, \ldots, x_n\}$ where $x_i \in \mathcal{X}$ and a function $f : \mathcal{X} \to \mathbb{R}$, we define the empirical expectation over the dataset as $\mathbb{E}_{\mathcal{D}}[f] = \frac{1}{n} \sum_{i=1}^{n} f(x_i)$. See additional notations in Appendix A.

| Entities | Forward | Backward | Direct | Indirect | Others |
|---|---|---|---|---|---|
| A, B, C, $A_i$, $B_i$, $C_i$ | $\rightarrow$ | $\leftarrow$ | $\rightarrow$ | $\rightsquigarrow$ | $R_1$, $R_2$ |

Table 1: Notations for tokens in Section 4. "$\rightarrow$" and "$\leftarrow$" denote forward and backward relationships for the reversal curse. "$\rightarrow$" and "$\rightsquigarrow$" denote direct and indirect implication for COT. $R_1$ and $R_2$ are relationship tokens in Section 4.3. A, B, C, $A_i$, $B_i$, $C_i$ denote tokens representing entities.

**Auto-regressive models.** Define the vocabulary $\mathcal{V} = [M]$ for a positive integer $M > 0$ which is the size of the vocabulary. Let $x = (x_1, x_2, \ldots, x_T)$ be a sequence of tokens of length $T$ where each token $x_t \in \mathcal{V}$, $\forall t \in [T]$. See Table 1 for notations of different tokens used in Section 4. We study auto-regressive models $p_\theta(\cdot|x) \in \Delta(\mathcal{V})$ parameterized by $\theta$ that take the sequence $x$ as input and predict the distribution of the next token $x_{T+1} \in \mathcal{V}$. For both models that we study in this paper, the next token probability is modeled as the softmax applied to the logits $l_\theta(\cdot|x) \in \mathbb{R}^M$ of each token in the vocabulary, i.e., $p_\theta(y|x) = \frac{\exp(l_\theta(y|x))}{\sum_{v \in \mathcal{V}} \exp(l_\theta(v|x))}$, $\forall y \in \mathcal{V}$. Also, each token $v \in \mathcal{V}$ has a corresponding (fixed or learnable) embedding vector $\boldsymbol{u}_v \in \mathbb{R}^d$.

## 3 Bilinear Models

We start analyzing the reversal curse under bilinear models, which can be viewed as simplified one-layer transformers with input length one and decoder layer only. Also, in this section, we assume the embeddings of each token are fixed, so we directly use the embedding vector to represent a token.

**Datasets.** Assume the vocabulary has size $m$ where each token $v_1, v_2, \ldots, v_m \overset{i.i.d.}{\sim} \mathcal{N}_d(0_d, \frac{1}{d}I_d)$. Let $\mathcal{V} = \{v_1, \ldots, v_m\}$ and let $\mathcal{X} = \{x_1, \ldots, x_n\}$ and $\mathcal{Y} = \{y_1, \ldots, y_n\}$ be disjoint random subsets of $\mathcal{V}$. Assume all training and test sequences have a length of two. For any $2 \leq i \leq n$, the training dataset contains both sequence $(x_i, y_i)$ and $(y_i, x_i)$. In addition, the training set contains $(x_1, y_1)$ while the test set only contain one example $(y_1, x_1)$. During training, the model learns both $(x_i, y_i)$ and $(y_i, x_i)$ for $i \geq 2$ to conclude that $(x_i, y_i)$ is equivalent to $(y_i, x_i)$. For example, $\mathcal{X}$ is a set of names and $\mathcal{Y}$ is a set of books. The sequence $(x_i, y_i)$ means "$x_i$ is the author of $y_i$", and the sentence $(y_i, x_i)$ means "$y_i$ is written by $x_i$". We test whether the model is able to infer an unseen sequence $(y_1, x_1)$ given the training data which includes the other direction $(x_1, y_1)$.

**Bilinear model.** We consider a bilinear model parameterized by $\Theta \in \mathbb{R}^{d \times d}$ of which the input contains single token $x \in \mathcal{V}$. The logits of the next token $y \in \mathcal{V}$ is defined as $l_\Theta(y|x) = x^\top \Theta y$ which is bilinear in $x$ and $y$, and thus the next token probability is $p_\Theta(y|x) = \frac{\exp(l_\Theta(y|x))}{\sum_{v \in \mathcal{V}} \exp(l_\Theta(v|x))}$. The training loss for the bilinear model is the cross-entropy loss $\mathcal{L}(\Theta) = \frac{1}{2n-1} \left( \sum_{i=1}^n -\log p_\Theta(y_i|x_i) + \sum_{i=2}^n -\log p_\Theta(x_i|y_i) \right)$ and the test loss (reversal loss) is $\mathcal{L}^{\texttt{rev}}(\Theta) = -\log p_\Theta(x_1|y_1)$. We study the training dynamics of gradient flow $\frac{d\Theta_t}{dt} = -\nabla \mathcal{L}(\Theta_t)$ with the initialization $\Theta_0$ that can be either randomly sampled from $\mathcal{N}(\mathbf{0}^{\otimes 2}, \sigma^2 I^{\otimes 2})$ or set as a pretrained parameter satisfying $\frac{1}{2m} < p_{\Theta_0}(y_i|x_i), p_{\Theta_0}(x_i|y_i) < \frac{2}{m}$ for all $i \in [n]$. The following theorem shows a separation during training dynamics.

**Theorem 1** (Separation of training dynamics (informal statement of Theorem 5)). *Fix any $\delta, \epsilon \in (0, 1)$. For small $\sigma$ and $d \geq \mathrm{poly}(n, m, 1/\epsilon, \log(1/\delta))$, with probability at least $1 - \delta$, we have*

$$\mathcal{L}^{\texttt{rev}}(\Theta_t)/\mathcal{L}^{\texttt{rev}}(\Theta_0) \geq (\mathcal{L}(\Theta_t)/\mathcal{L}(\Theta_0))^\epsilon, \ \forall t \geq 0.$$

Theorem 1 shows that the reversal loss is lower bounded by the training loss. Note that for large $d$ and small $\epsilon$ close to 0, $(\mathcal{L}(\Theta_t)/\mathcal{L}(\Theta_0))^\epsilon$ is close to 1 and thus $\mathcal{L}^{\texttt{rev}}(\Theta_t) \gtrsim \mathcal{L}^{\texttt{rev}}(\Theta_0)$ which implies that $p_\Theta(x_1|y_1)$ remains small during training. We summarize the above argument in Theorem 2.

**Theorem 2** (Lower bound of reversal loss (informal statement of Theorem 6). *Fix arbitrary $c > 0$ and $C \leq \log(m/2)$. Suppose $\sigma$ is small and $d \geq \mathrm{poly}(n, m, \log(1/\delta), \log c, 1/\log C)$. With probability at least $1 - \delta$, it holds that $\mathcal{L}^{\texttt{rev}}(\Theta_\tau) \geq C$, where $\tau$ denotes the first time such that $\mathcal{L}(\Theta_t) \leq c$.*

The proofs of Theorems 1 and 2 are deferred to Appendix B. Theorem 2 implies that for large $d^2$, while the training loss can be trained to be arbitrarily small, the reversal loss remains large. In other words, the model fails to infer an unseen sequence $(y_1, x_1)$ given the training data which includes the other direction $(x_1, y_1)$. Furthermore, even if the model is fine-tuned on new data from a pre-trained parameter $\Theta_0$ that initially grasps the concept of reversal and satisfies $\frac{1}{2m} < p_{\Theta_0}(y_i|x_i), p_{\Theta_0}(x_i|y_i) < \frac{2}{m}$ for new data, it fails to extend this understanding to new, unseen data.

A core reason that the reversal curse happens on the above bilinear model is that the parameter matrix $\Theta_t$ is asymmetric. Consequently, the logits $l_{\Theta_t}(y|x) = x^\top \Theta_t y$ and $l_{\Theta_t}(x|y) = y^\top \Theta_t x$ generally differ. Consider a special case where each $v_i$ is a one-hot vector. Then $l_\Theta(y|x) = x^\top \Theta y = \Theta_{ij}$ and $l_\Theta(x|y) = y^\top \Theta x = \Theta_{ji}$ for $x = e_i, y = e_j$. Training on $(x, y)$ can increase $\Theta_{ij}$ but not $\Theta_{ji}$, which means the model does not automatically learn the reversal direction $(y, x)$ from $(x, y)$. In Section 4, we will show that for one-layer transformers, the reversal curse is mainly caused by the same reason, i.e., the asymmetry of the model weights.

## 4 One-Layer Transformers

In Section 3, we analyzed the reversal curse under a bilinear model. In this section, we analyze the reversal curse for one-layer transformers in a similar setting to [14] via training dynamics. We also extend our analysis to chain-of-thought in Section 4.2.

**Basic notations.** Let $\mathcal{V} = [M]$ be the vocabulary. For any token $x \in [M]$, we also use the corresponding one-hot vector $\boldsymbol{x} = \boldsymbol{e}_x \in \mathbb{R}^M$ to represent it. Let $U = [\boldsymbol{u}_1, \boldsymbol{u}_2, \dots, \boldsymbol{u}_M]^\top \in \mathbb{R}^{M \times d}$ be the embedding matrix, where $\boldsymbol{u}_x \in \mathbb{R}^d$ is the embedding of token $x \in [M]$. Note that $U^\top \boldsymbol{x} = \boldsymbol{u}_x$. Consider the $i$-th training sample in the dataset, $x[i] = (x_1[i], \dots, x_{T[i]}[i], x_{T[i]+1}[i])$, a sequence of tokens of length $T[i] + 1$. Here, $x_{T[i]+1}[i]$ is the next token (or equivalently the label) to be predicted, $x_{T[i]}[i]$ is the query token, and $(x_1[i], \dots, x_{T[i]-1}[i])$ are contextual tokens. For token $x_t[i]$, we also use its one-hot vector $\boldsymbol{x}_t[i] = \boldsymbol{e}_{x_t[i]} \in \mathbb{R}^M$ to represent it. Define the contextual token matrix $X[i] = [\boldsymbol{x}_1[i], \dots, \boldsymbol{x}_{T[i]}[i]]^\top \in \mathbb{R}^{(T[i]-1) \times M}$. We omit all $i$ in notations when the context is clear.

**One-layer transformer.** For a training sample $x = (x_1, \dots, x_T, x_{T+1})$, its contextual token matrix $X = [\boldsymbol{x}_1, \dots, \boldsymbol{x}_{T-1}]^\top$ and thus $XU = [\boldsymbol{u}_{x_1}, \dots, \boldsymbol{u}_{x_{T-1}}]^\top$ contains the contextual token embeddings. We study one-layer transformers in the same setting as [14]. In particular, for an input token sequence $(x_1, \dots, x_T)$, after the one-layer self-attention, we can obtain $\tilde{\boldsymbol{u}}_T = U^\top \text{LN}(X^\top \boldsymbol{b}_T)$ where $b_{tT} = \frac{\exp(\boldsymbol{u}_{x_T}^\top W_Q W_K^\top \boldsymbol{u}_{x_t}/\sqrt{d})}{\sum_{t'=1}^{T-1} \exp(\boldsymbol{u}_{x_T}^\top W_Q W_K^\top \boldsymbol{u}_{x_{t'}}/\sqrt{d})}$, $\boldsymbol{b}_T = [b_{1T}, \dots, b_{T-1,T}]^\top$ contains attention scores (after softmax) that query token $x_T$ attend to each contextual token[3], $\text{LN}(\boldsymbol{x}) = \boldsymbol{x}/\|\boldsymbol{x}\|_2$ is the $\ell_2$-normalization operator, and $W_Q, W_K \in \mathbb{R}^{d \times d_k}$ are trainable query and key matrices respectively. The logit of $x \in [M]$ is then calculated by a decoder layer, i.e., $l_\theta(x|x_1, \dots, x_T) = \boldsymbol{u}_x^\top W_V \tilde{\boldsymbol{u}}_T$, where $\theta$ encodes all parameters in the transformer, and $W_V \in \mathbb{R}^{d \times d}$ can be viewed as a reparameterization of value and output matrices. Finally, the next token prediction probability is obtained by

$$p_\theta(x|x_1, \dots, x_T) = \frac{\exp(l_\theta(x|x_1, \dots, x_T))}{\sum_{x' \in [M]} \exp(l_\theta(x|x_1, \dots, x_T))} = \frac{\exp(\boldsymbol{u}_x^\top W_V \tilde{\boldsymbol{u}}_T)}{\sum_{x' \in [M]} \exp(\boldsymbol{u}_{x'}^\top W_V \tilde{\boldsymbol{u}}_T)}.$$

We use the cross-entropy loss function to train the model over the whole training set $\mathcal{D}_{\text{train}}$, i.e., $\max_{U, W_K, W_Q, W_V} J \triangleq \mathbb{E}_{\mathcal{D}_{\text{train}}}[\log p_\theta(x_{T+1}|x_1, \dots, x_T)]$.

**Reparameterization.** Similar to [14], we define $Z = UW_Q W_K^\top U^\top / \sqrt{d} \in \mathbb{R}^{M \times M}$ and $Y = UW_V^\top U^\top \in \mathbb{R}^{M \times M}$ and are interested in their dynamics after reparameterization. Then, the attention score (after softmax) and next token probability become

$$b_{tT} = \frac{\exp(\boldsymbol{x}_T^\top Z \boldsymbol{x}_t)}{\sum_{t'=1}^{T-1} \exp(\boldsymbol{x}_T^\top Z \boldsymbol{x}_{t'})}, \quad p_\theta(x|x_1, \dots, x_T) = \frac{\exp\left(\boldsymbol{x}^\top Y^\top \text{LN}(X^\top \boldsymbol{b}_T)\right)}{\sum_{x'} \exp\left(\boldsymbol{x'}^\top Y^\top \text{LN}(X^\top \boldsymbol{b}_T)\right)}, \quad (1)$$

---

[2]Empirically, $d$ only needs to be of the order of logarithm of the vocabulary size. See Appendices E.2.1 and E.2.2 for additional results.

[3]Note that we assume the query token will not attend to itself as in [14].

and the objective can be written as

$$\max_{Y,Z} J = \mathbb{E}_{\mathcal{D}_{\text{train}}}[\boldsymbol{x}_{T+1}^\top Y^\top \text{LN}(X^\top \boldsymbol{b}_T) - \log \sum_{x' \in [M]} \boldsymbol{x'}^\top Y^\top \text{LN}(X^\top \boldsymbol{b}_T)]. \tag{2}$$

Let $\eta_Y$, $\eta_Z$ be the learning rate of matrices $Y$ and $Z$ respectively. Then the gradient of $Y$ and $Z$ can be characterized by the following lemma:

**Lemma 1** (Gradient of $Y$ and $Z$ for 1-layer transformer, Lemma 1 of [14])**.** *The gradient of $Y$ and $Z$ w.r.t.* (2) *of batch size 1 and learning rate $\eta_Y$ and $\eta_Z$ can be written as*

$$\dot{Y} = \eta_Y LN(X^\top \boldsymbol{b}_T)(\boldsymbol{x}_{T+1} - \boldsymbol{\alpha})^\top, \quad \dot{Z} = \eta_Z \boldsymbol{x}_T(\boldsymbol{x}_{T+1} - \boldsymbol{\alpha})^\top Y^\top \frac{P_{X^\top \boldsymbol{b}_T}^\perp}{\|X^\top \boldsymbol{b}_T\|_2} X^\top \text{diag}(\boldsymbol{b}_T)X, \tag{3}$$

*where $P_{\boldsymbol{v}}^\perp \triangleq I - \boldsymbol{v}\boldsymbol{v}^\top/\|\boldsymbol{v}\|_2^2$ projects any vector to orthogonal complement of $\boldsymbol{v}$, $\boldsymbol{\alpha} = [\alpha_1, \alpha_2, \ldots, \alpha_M]^\top \in \mathbb{R}^M$ with $\boldsymbol{\alpha} = \exp\left(Y^\top LN(X^\top \boldsymbol{b}_T)\right)/\mathbf{1}^\top \exp\left(Y^\top LN(X^\top \boldsymbol{b}_T)\right)$.*

*Proof.* $\dot{Y}, \dot{Z}$ can be obtained by direct calculation. One can refer to the proof of Lemma 1 of [14]. □

### 4.1 Main results for the reversal curse

In this section, we analyze the reversal curse where data points are three-token sentences "A $\to$ B" or "B $\leftarrow$ A". For each sentence, A and B are two distinct tokens that represent two entities, and "$\to$" and "$\leftarrow$" are two special tokens representing a pair of relationships inverse to each other.

**Datasets.** Let $N_{\text{train}} > 0$, $N_{\text{test}}^{(1)} > 0$ and $N_{\text{test}}^{(2)} > 0$ and denote $N_{\text{total}} = N_{\text{train}} + N_{\text{test}}^{(1)} + N_{\text{test}}^{(2)}$. Let $A_i, B_i \in \mathcal{V}, \forall i \in [N_{\text{total}}]$ be $2N_{\text{total}}$ distinct tokens representing distinct entities. Let $\to, \leftarrow \in \mathcal{V}$ be two additional different tokens that represent two inverse relationships. Specifically, we have $A_i \to B_i$ and $B_i \leftarrow A_i$ for all $i \in [N_{\text{total}}]$. For notation convenience, we define the following three index sets

$$\mathcal{I}_{\text{train}} = [N_{\text{train}}], \qquad \mathcal{I}_{\text{test}}^{(1)} = [N_{\text{train}} + N_{\text{test}}^{(1)}]\backslash\mathcal{I}_{\text{train}}, \qquad \mathcal{I}_{\text{test}}^{(2)} = [N_{\text{total}}]\backslash(\mathcal{I}_{\text{train}} \cup \mathcal{I}_{\text{test}}^{(1)}).$$

The training set $\mathcal{D}_{\text{train}}$ consists of all $A_i \to B_i$ and $B_i \leftarrow A_i$ for $i \in \mathcal{I}_{\text{train}}$. In addition, $\mathcal{D}_{\text{train}}$ contains $A_i \to B_i$ for $i \in \mathcal{I}_{\text{test}}^{(1)}$ and $B_i \leftarrow A_i$ for $i \in \mathcal{I}_{\text{test}}^{(2)}$. For convenience, we let $N = |\mathcal{D}_{\text{train}}|$ to be the size of the training set. The test set $\mathcal{D}_{\text{test}}$ consists of $B_i \leftarrow A_i$ for $i \in \mathcal{I}_{\text{test}}^{(1)}$ and $A_i \to B_i$ for $i \in \mathcal{I}_{\text{test}}^{(2)}$. Under our construction of the dataset, the LLM will learn the relationship between $A_i$ and $B_i$ for $i \in \mathcal{I}_{\text{train}}$ in both directions to deduce that $\to$ is reverse to $\leftarrow$, and learn the relationship between $A_i$ and $B_i$ for $i \in \mathcal{I}_{\text{test}}^{(1)} \cup \mathcal{I}_{\text{test}}^{(2)}$ in one direction and will be tested for the other.

We use $p_\theta(A_i|B_i \leftarrow)$ and $p_\theta(B_i|A_i \to)$ to more compactly represent $p_\theta(x_3 = A_i|x_1 = B_i, x_2 =\leftarrow)$ and $p_\theta(x_3 = B_i|x_1 = A_i, x_2 =\to)$, respectively. Our goal is to prove through the training dynamics of one-layer transformers that the test probability remains negligible during training. In particular, we are interested in $p_\theta(A_i|B_i \leftarrow), \forall i \in \mathcal{I}_{\text{test}}^{(1)}$ and $p_\theta(B_i|A_i \to), \forall i \in \mathcal{I}_{\text{test}}^{(2)}$.

For convenience, we assume zero-initialization $Y(0) = \mathbf{0}$ and $Z(0) = \mathbf{0}$. This is the same as [14] and is reasonable since empirically, $Y$ and $Z$ are usually initialized as inner products of $d$-dimensional vectors with i.i.d Gaussian entries, and thus are almost zero (Lemma 8 in Appendix B). The following proposition shows the initial train/test probabilities are uniform over the vocabulary $\mathcal{V}$.

**Proposition 4.1** (Initial probability under zero initializaion)**.** *Assume the transformer is under zero-initialization $\theta(0) = (Y(0), Z(0))$ with $Y(0) = \mathbf{0}$ and $Z(0) = \mathbf{0}$. For any $i \in [N_{total}]$, we have*

$$p_{\theta(0)}(B_i|A_i \to) = p_{\theta(0)}(A_i|B_i \leftarrow) = 1/M.$$

The proof is deferred to Appendix C.1.1. Proposition 4.1 shows that initially, the probability of predicting any B (or A, respectively) given any A $\to$ (or B $\leftarrow$, respectively) as input is uniform over the whole vocabulary. When $Y(0)$ and $Z(0)$ are not exactly $\mathbf{0}$ but close to $\mathbf{0}$, the initial prediction will still be close to the uniform distribution, which is similar to Lemma 6. Next we analyze the dynamics of $p_{\theta(t)}(B_i|A_i \to)$ and $p_{\theta(t)}(A_i|B_i \leftarrow)$.

**Proposition 4.2** (Next token probability)**.** *For input sequence $(x_1, x_2)$, the next token probability under parameters $\theta(t)$ is $p_{\theta(t)}(x|x_1, x_2) = \exp\left(Y(t)_{x_1,x}\right)/\sum_{x' \in [M]} \exp\left(Y(t)_{x_1,x'}\right)$, where $Y(t)_{i,j}$ is the entry of the matrix $Y(t)$ at row $i$ and column $j$.*

The proof is also deferred to Appendix C.1.1. According to Proposition 4.2, the next token probability when the entity in the input is $x_1$ is determined by the $x_1$-th row of the matrix $Y(t)$. Another nice property indicated by Proposition 4.2 is that we don't need to keep track of the dynamics of $Z(t)$, which could greatly simplify the analysis. The following lemma shows the dynamics of $Y(t)$.

**Lemma 2** (Dynamics of $Y(t)$). *Assume we run SGD with batch size 1 [4], and assume $M \gg 100$ and $\frac{1}{M^{0.99}} \ll \eta_Y < 1$. Let $t \gtrsim \frac{N \ln M}{\eta_Y}$ and let $Y(t)_i$ denote the $i$-th row of $Y(t)$ and $Y(t)_{ij}$ denote the $(i, j)$-th entry of $Y(t)$. Then for training sequence $(x_1, x_2, x_3) \in \mathcal{D}_{train}$ at time $t$, we have*

$$Y(t)_{x_1, x_3} \gtrsim \ln\left(M \eta_Y t/N\right), \quad \text{and} \quad Y(t)_{x_1, x} \lesssim -\ln\left(M \eta_Y t/N\right)/M, \quad \forall x \neq x_3,$$

*and for any test sequence $(x_1, x_2, x_3) \in \mathcal{D}_{test}$, we have $Y(t)_{x_1, x} = 0, \forall x \in [M]$.*

The proof of Lemma 2 is presented in Appendix C.1.2. Lemma 2 implies the asymmetry of the model weights $Y(t)$: for two tokens $x_1, x_3$, when $x_1$ appears as a contextual token and $x_3$ serves as the next token in the same training sequence, the model weights $Y(t)_{x_1, x_3}$ gets increased during training while $Y(t)_{x_3, x_1}$ will not get increased. Combining Proposition 4.2, we can obtain our main theorem for the reversal curse.

**Theorem 3** (Reversal curse). *Assume we run SGD with batch size 1, and assume $M \gg 100$ and $\frac{1}{M^{0.99}} \ll \eta_Y < 1$. Let $t \gtrsim \frac{N \ln M}{\eta_Y}$ denote the time step which also satisfies $\ln t \gtrsim \ln(NM/\eta_Y)$. For training sequence $(x_1, x_2, x_3) \in \mathcal{D}_{train}$ at time $t$, we have*

$$p_{\theta(t)}(x_3|x_1, x_2) \geq 1 - (M-1)(M\eta_Y t/N)^{-c} \to 1, \quad \text{as } t \to \infty$$

*for some constant $c > 0$, and for any test sequence $(x_1, x_2, x_3) \in \mathcal{D}_{test}$ that is not included in the training set $\mathcal{D}_{train}$, we have $p_{\theta(t)}(x_3|x_1, x_2) \leq 1/M$.*

Theorem 3 shows that although the direction presented in the training set can be learned nearly perfectly, the model's next token prediction of the reverse direction is almost a random guess. The proof is deferred to Appendix C.1.3. We also empirically validate the above results for multi-layer transformers in Section 5.

## 4.2 Chain-of-thought

In this section, we extend our analysis in Section 4.1 to study other logical relationships. In particular, we study chain-of-thought (COT) [4] and show its importance via training dynamics. COT encourages LLMs to output a series of intermediate reasoning steps to increase their performance. Consider the simplest example, where the model learns two facts that $A \to B$ and $B \to C$, and we want to test whether the model is able to directly conclude that $A \rightsquigarrow C$. COT indicates that if an LLM is only trained on $A \to B$ and $B \to C$, it would be easier for the model to deduce $A \rightsquigarrow C$ during the inference time if the model can first output the intermediate steps $A \to B$ and $B \to C$, instead of directly predicting the next token $C$ given the input "$A \rightsquigarrow$". The failure of directly deducing $A \rightsquigarrow C$ is also empirically observed by [9].

Theoretically, [15] shows the importance of COT for some complex reasoning tasks through the lens of the expressivity of transformers. In this section, we show the importance of COT through a different angle, i.e., training dynamics. We show that for the above simplest two-step reasoning, without COT, the model is not able to directly predict $C$ given the input "$A \rightsquigarrow$" even if it learns $A \to B$ and $B \to C$.

**Theorem 4** (Importance of chain-of-thought, informal statement of Theorem 7). *Under certain assumptions as stated in Theorem 7, for any $A_i, B_i, C_i$ s.t. $A_i \to B_i$ and $B_i \to C_i$ are in the training set but $A_i \rightsquigarrow C_i$ is not, we have*

$$p_{\theta(t)}(B_i|A_i \to) \to 1, \quad p_{\theta(t)}(C_i|B_i \to) \to 1, \quad p_{\theta(t)}(C_i|A_i \rightsquigarrow) \leq 1/M, \quad \text{as } t \to \infty.$$

We defer the details of the dataset construction and proof to Appendix C.2. Theorem 4 shows that although the LLM learns $A_i \to B_i$ and $B_i \to C_i$ nearly perfectly, it cannot directly deduce $A_i \rightsquigarrow C_i$. Analogous to the asymmetry of causal transformer weights as we discussed in Section 4.1, our analysis of COT reveals another property, i.e., intransitivity: training the weights associated with $A$ to $B$ and $B$ to $C$ does not necessarily increase the weights associated with $A$ to $C$.

---

[4]The lemma holds even if the batch size is larger than 1 and the analysis is essentially the same.

We also emphasize that the model fails to directly deduce $A_i \rightsquigarrow C_i$ when the two intermediate steps $A_i \rightarrow B_i$ and $B_i \rightarrow C_i$ are trained *separately*. If the two steps are concatenated into a single training sequence, it is possible that the model learns $A_i \rightsquigarrow C_i$ directly [19].

### 4.3    Roles of the attention score matrix

During the analysis of Sections 4.1 and 4.2, we show that the reversal curse and the importance of COT are largely due to the asymmetry and intransitivity of causal transformer weights (in our case, the weight matrix $Y(t)$). However, it seems that the dynamics of the attention score matrix $Z(t)$ do not impact the model performance. Below, we briefly discuss the role of the attention score matrix $Z(t)$.

In (1), the attention score is used to calculate the weights $b_{tT}$, where a contextual token $x_t$ with a larger attention score attended by the query token $x_T$ has a larger weight. Note that we use the same formulation as the previous work [14] where the query token will not attend to itself. Therefore, for a three-token training sequence, the weights $b_{12}$ is always one since there is only one contextual token $x_1$, no matter whether the value of the attention score is high or low.

However, consider a slightly different setting, where the relationship is represented by two tokens. In that case, $x_1 = A_i, x_2 = R_1, x_3 = R_2, x_4 = B_i$, and there are two contextual tokens $A_i$ and $R_1$. The role of the attention score is then to select the important token, i.e., $A_i$, by putting more weights on it. Theorem 2 of [14] showed that under certain assumptions, the query token $R_2$ will attend more to "distinct tokens" $A_i$ and less to the "common token" $R_1$. Therefore, the query token $R_2$ will eventually put all weights to $A_i$, and the remaining analysis remains the same as in Sections 4.1 and 4.2. See Appendix C.3 for a more rigorous analysis.

## 5    Experiments

In this section, we conduct experiments to further validate our theoretical results in Section 4 on multi-layer transformers. We show experimental results of the reversal curse in this section and COT in Appendix D. Note that in Sections 3 and 4, we theoretically proved the reversal curse for both the bilinear model and one-layer transformer under certain assumptions. Now, we empirically show that the reversal curse still happens even for multi-layer transformers. In Appendix E.2.3, we also provide empirical results that the reversal curse does not happen in ICL settings.

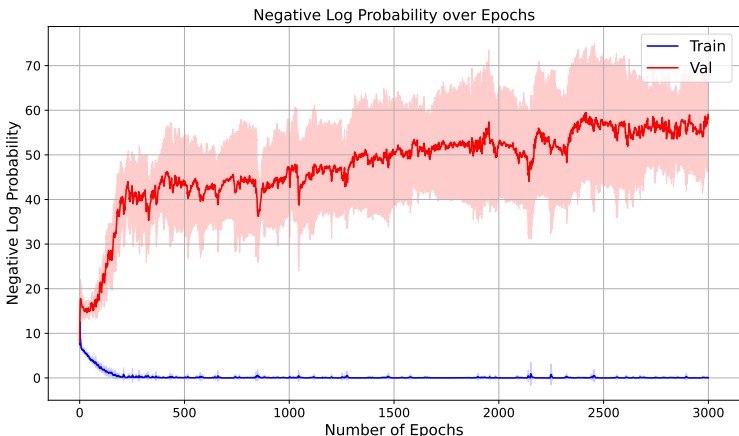

Figure 1: Experiment results of reversal curse under default configuration (see Table 3). The curves represent the (average) negative log probability of the model predicting the next token to be $B_i$ when the input is "$A_i \rightarrow$", or to be $A_i$ when the input is "$B_i \leftarrow$". While the sentences in the training set can be learned nearly perfectly (as shown by the training curve where the next token probability converges to one), the model is not able to predict the correct next token in the validation set better than a uniformly random guess. Both curves are averaged over 10 random seeds.

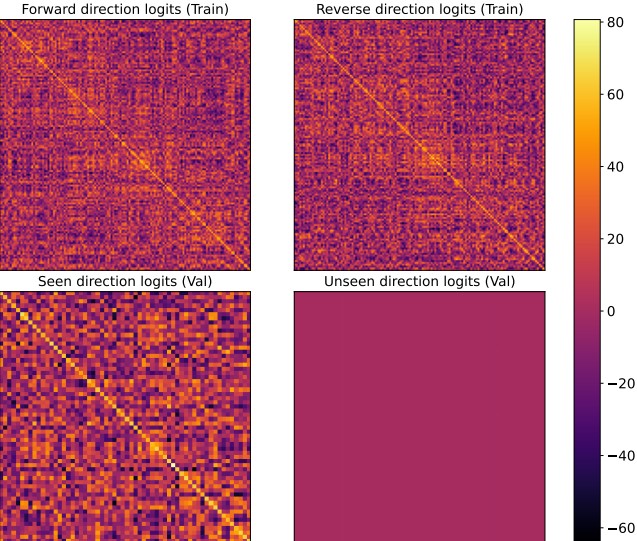

Figure 2: Visualization of the weights (logits) of the model with default configurations trained after 3000 epochs for the reversal curse experiment. For the top-left matrix, the $i$-th row corresponds to an entity token $A_i$ for a training pair, and the $i$-th column corresponds to an entity token $B_i$ for a training pair. The $(i, j)$-th entry represents the model weights from the token $A_i$ to $B_j$, i.e., the logits of $B_j$ when the input sequence consists of only $A_i$. Similarly, for the bottom-left matrix, the row corresponds to the input entity tokens of the seen direction (the direction included in the training set) of validation pairs, and the column corresponds to output entity tokens. The two matrices on the right are obtained by swapping row tokens and column tokens of their corresponding left matrices. Note that the diagonals of the bottom-right matrix are all close to zero, while the diagonals of other matrices all have large values. This implies that if a pair of tokens $(A, B)$ only appear in the training set in one direction, then the model weights associated with the other direction will hardly get trained.

**Dataset construction.** Below, we describe how we generate our synthetic dataset for experiments on the reversal curse. We choose the vocabulary $\mathcal{V} = \{0, 1, \ldots, N\}$ for a specified $N > 0$. We randomly sample two disjoint sets of entities $\mathcal{A}, \mathcal{B} \subset \mathcal{V}$ with $|\mathcal{A}| = |\mathcal{B}| = |\mathcal{V}|/4$, and reserve two additional tokens for relationships $\rightarrow$ and $\leftarrow$, respectively. [5] Next, we specify a bijection from $\mathcal{A}$ to $\mathcal{B}$ uniformly at random. For each $A_i \in \mathcal{A}$ and its corresponding $B_i \in \mathcal{B}$, we can obtain a pair of sequence $(A_i \rightarrow B_i, B_i \leftarrow A_i)$. We split the set of all pairs into training pairs and validation pairs. For each training pair, both sequences will be included in the training set, while for the validation pair, we randomly select one sequence for the training set and the other for the validation set. Therefore, the model will learn both directions for the training pairs and only one direction for each validation pair while being tested in the unseen direction.

**Model architectures.** We train multi-layer transformers based on GPT-2 architecture [63]. Figure 1 shows the results where the model has 24 layers, 12 attention heads per layer, uses absolute positional encoding, and we choose the vocabulary size of 800. The training set size is 340, and the validation set size is 60 (resulting from 140 training pairs and 60 validation pairs). We also conducted experiments with various model configurations and vocabulary sizes in Appendix E.2. Besides, all hyperparameters and different model configurations are presented in Appendix E.1.

**Results.** Figure 1 shows that during the training, the next token probability for training data increases a lot while the next token probability for validation data remains unchanged or gets even smaller. This is consistent with our theoretical results of Theorem 3.

According to our theoretical analysis, the reversal curse happens due to the asymmetry of model (re-parameterized) weights (i.e., logits of a token given another token as input), and we also empirically validate the asymmetry for multi-layer transformers. Figure 2 shows the model weights from a token $x_1$ to $x_3$ is trained large for a training sequence $(x_1, x_2, x_3)$ as represented by the diagonals of the

---

[5]By default, each entity consists of one token. See multi-token experiments in Appendices E.2 and E.3

first three matrices, while the weights from a token $x_1$ to $x_3$ remains nearly zero for a validation sequence $(x_1, x_2, x_3)$ as represented by the diagonals of the last matrix, which is consistent with Lemma 2. This implies that if a pair of tokens $(\mathtt{A}, \mathtt{B})$ only appear in the training set in one direction, then the model weights associated with the other direction will hardly get trained.

## 6 Conclusions

In this paper, we study the reversal curse theoretically via training dynamics of (1) a bilinear model, which is a simplification of the one-layer transformer; (2) one-layer transformers under certain technical assumptions similar to [14]. Our theoretical results suggest that a core reason the reversal curse happens in auto-regressive LLMs is the asymmetry of the model weights, and we apply our technique to prove the necessity of COT for one-layer transformers, which is mainly due to the intransitivity of model weights. The asymmetry and intransitivity of model weights caused by unconstrained optimization of CE loss indicate that an auto-regressive LLM might mainly focus on learning text sequences during training *separately* instead of automatically deducing indirect conclusions under the current popular training paradigms. This highlights the importance of ICL, data augmentation, or planning for current auto-regressive LLMs to solve complex reasoning tasks.

As for future directions, it would be interesting and important to study: (1) What is a unified way to characterize and study the reversal curse, COT, and other similar logical reasoning tasks? (2) Our paper mainly focuses on three-token sequences, where each entity or relationship is represented by a single token. While we empirically explored the setting where each entity might consist of multiple tokens and distinct entities might share a few tokens, it would be interesting to analyze the multiple-token setting theoretically. (3) We theoretically analyzed the bilinear model and one-layer transformer, and it would be an important future direction to extend the analysis to multi-layer transformers.

## Acknowledgements

This work was partially supported by a gift from Open Philanthropy to the Center for Human-Compatible AI (CHAI) at UC Berkeley and by NSF Grants IIS-1901252 and CCF-2211209. The work was done when HZ was a visiting researcher at Meta.

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

## A   Additional Notations

Let $\delta_{ij} = 1$ for $i = j$ and $\delta_{ij} = 0$ for $i \neq j$. For a squared matrix $A \in \mathbb{R}^{d \times d}$, its trace is $\text{Tr}(A) = \sum_{i=1}^{d} A_{ii}$. For two matrices $A, B \in \mathbb{R}^{m \times n}$ of the same shape, their inner product is defined as $\langle A, B \rangle = \text{Tr}(AB^\top)$. For any matrix $A \in \mathbb{R}^{m \times n}$, its (Frobenius) norm is defined as $\|A\| = \sqrt{\langle A, A \rangle}$. For any vector $\boldsymbol{x} = (x_1, \ldots, x_d)^\top \in \mathbb{R}^d$ or a matrix $A \in \mathbb{R}^{m \times n}$, we define the zero-norm as $\|\boldsymbol{x}\|_0 = \sum_{i=1}^{d} \mathbb{1}\{x_i \neq 0\}$ or $\|A\|_0 = \sum_{i=1}^{m} \sum_{j=1}^{n} \mathbb{1}\{A_{ij} \neq 0\}$ where $\mathbb{1}\{\cdot\}$ is the indicator function.

## B   Missing Proofs of Section 3

**Theorem 5** (Separation of training dynamics, formal statement of Theorem 1). *Fix arbitrary $\delta, \epsilon \in (0, 1)$. Let $v_1, \ldots, v_m$ be independently sampled from $\mathcal{N}_d(\mathbf{0}_d, \frac{1}{d} I_d)$. Let $x_1, \ldots, x_n$ and $y_1, \ldots, y_n$ be sampled uniformly at random from $\{v_1, \ldots, v_m\}$ without replacement. Define*

$$\mathcal{L}(\Theta) = \frac{1}{2n-1} \left( \sum_{i=1}^{n} -\log p_\Theta(y_i|x_i) + \sum_{i=2}^{n} -\log p_\Theta(x_i|y_i) \right)$$

$$\mathcal{L}^{rev}(\Theta) = -\log p_\Theta(x_1|y_1).$$

*Consider the gradient flow $\Theta_t : t \geq 0$*

$$\frac{d\Theta_t}{dt} = -\nabla \mathcal{L}(\Theta_t).$$

*where $\Theta_0 \sim \mathcal{N}(\mathbf{0}^{\otimes 2}, \sigma^2 \cdot I^{\otimes 2})$ or $\Theta_0$ satisfies $\frac{1}{2m} < p_{\Theta_0}(y_i|x_i), p_{\Theta_0}(x_i|y_i) < \frac{2}{m}$ for all $i \in [n]$. Suppose $\sigma \leq \frac{1}{100 \ln(64m^2/\delta)}$ and*

$$d \geq \frac{10^6 n^4 m^2 \log^4(2m) \log(64m^2 n^2/\delta)}{\epsilon^2}.$$

*With probability at least $1 - \delta$, we have*

$$\frac{\mathcal{L}^{rev}(\Theta_t)}{\mathcal{L}^{rev}(\Theta_0)} \geq \left( \frac{\mathcal{L}(\Theta_t)}{\mathcal{L}(\Theta_0)} \right)^\epsilon, \quad \forall t \geq 0.$$

*Proof.* Let $v = \sqrt{\frac{400 n^2 m^2 \log(64m^2 n^2/\delta)}{d}}$. By Lemma 3 and Lemma 4, with probability at least $1 - \delta$,

$$\mathcal{L}^{\text{rev}}(\Theta_t) \geq \mathcal{L}^{\text{rev}}(\Theta_0) \cdot \left( 1 + \frac{\mathcal{L}(\Theta_0) \cdot t}{8(2n-1)\log^2(2m)} \right)^{-8v(2n-1)\log^2(2m)}.$$

$$\geq \mathcal{L}^{\text{rev}}(\Theta_0) \cdot \left( \frac{\mathcal{L}(\Theta_t)}{\mathcal{L}(\Theta_0)} \right)^{8v(2n-1)\log^2(2m)}$$

By definition of $d$, we have $8v(2n-1)\log^2(2m) \leq \epsilon$. Notice that $\frac{\mathcal{L}(\Theta_t)}{\mathcal{L}(\Theta_0)} < 1$, thus

$$\mathcal{L}^{\text{rev}}(\Theta_t) \geq \mathcal{L}^{\text{rev}}(\Theta_0) \cdot \left( \frac{\mathcal{L}(\Theta_t)}{\mathcal{L}(\Theta_0)} \right)^\epsilon.$$

$\square$

**Theorem 6** (Lower bound of reversal loss, formal statement of Theorem 2). *Fix arbitrary $c > 0$ and $C \leq \log(m/2)$. Under the setting of Theorem 5, suppose $\sigma \leq \frac{1}{100 \ln(64m^2/\delta)}$ and*

$$d \geq 10^6 n^4 m^2 \log^4(2m) \log(64m^2 n^2/\delta) \cdot \frac{\log^2 \frac{c}{\log(2m)}}{\log^2 \frac{C}{\log(m/2)}}.$$

*With probability at least $1 - \delta$,*

$$\mathcal{L}^{rev}(\Theta_\tau) \geq C.$$

*where $\tau$ denotes the first time such that $\mathcal{L}(\Theta_t) \leq c$.*

*Proof.* By continuity, $\mathcal{L}(\Theta_\tau) = c$. By Theorem 5, when

$$d \geq \frac{10^6 n^4 m^2 \log^4(2m) \log(64m^2 n^2/\delta)}{\epsilon^2} \tag{4}$$

with probability at least $1 - \delta$,

$$\mathcal{L}^{\text{rev}}(\Theta_\tau) \geq \mathcal{L}^{\text{rev}}(\Theta_0) \cdot \left( \frac{\mathcal{L}(\Theta_\tau)}{\mathcal{L}(\Theta_0)} \right)^\epsilon$$

$$\geq \mathcal{L}^{\text{rev}}(\Theta_0) \cdot \left( \frac{c}{\mathcal{L}(\Theta_0)} \right)^\epsilon.$$

Under this event, applying Lemma 6, we can obtain

$$\mathcal{L}^{\text{rev}}(\Theta_\tau) \geq \log(m/2) \cdot \left( \frac{c}{\log(2m)} \right)^\epsilon.$$

To ensure that the right hand side is $C$, we set $\epsilon = \frac{\log \frac{C}{\log(m/2)}}{\log \frac{c}{\log(2m)}}$. One may check that the definition of $d$ satisfies Eq. (4). It follows that

$$\mathcal{L}^{\text{rev}}(\Theta_\tau) \geq C.$$

$\square$

## B.1 Training dynamics

**Lemma 3** (Dynamics of the forward loss). *Let $x_1, \ldots, x_n$, $y_1, \ldots, y_n$, $\mathcal{L}(\Theta)$, and $\Theta_t$ $(t \geq 0)$ be defined as in Theorem 5. When $\sigma \leq \frac{1}{100 \ln(16n^2/\delta)}$ and $d \geq 1600n^3 m^2 \log(8m^2 n^2/\delta)$, with probability at least $1 - \delta$, we have*

$$\mathcal{L}(\Theta_t) \leq \frac{1}{\frac{t}{8(2n-1)\log^2(2m)} + \frac{1}{\mathcal{L}(\Theta_0)}}, \quad \forall t \geq 0.$$

*Furthermore,*

$$\inf \left\{ p_{\Theta_t}(x_i|y_i), p_{\Theta_t}(y_i|x_i) : t \geq 0, i \in [n] \right\} > \frac{1}{2m}.$$

*Proof.* For convenience, we assume $x_1, \ldots, x_n = v_1, \ldots, v_n$ and $y_1, \ldots, y_n = v_{n+1}, \ldots v_{2n}$ WLOG.

Let $\epsilon = \sqrt{\frac{400n^2 m^2 \log(8m^2 n^2/\delta)}{d}}$. Then $\epsilon \leq \frac{1}{2\sqrt{n}}$. Define $l_i(\Theta) = -\log p_\Theta(y_i|x_i)$ and $l_i^{\text{rev}}(\Theta) = -\log p_\Theta(x_i|y_i)$. Let $\alpha_{i,j}^{(t)} = -p_{\Theta_t}(v_j|v_i) + \delta_{i,j-n}$, $\beta_{i,j}^{(t)} = -p_{\Theta_t}(v_j|v_i) + \delta_{i-n,j}$. By Lemma 5,

$$\frac{d\mathcal{L}(\Theta_t)}{dt} = \left\langle \nabla \mathcal{L}(\Theta_t), \frac{d\Theta_t}{dt} \right\rangle = -\left\langle \nabla \mathcal{L}(\Theta_t), \nabla \mathcal{L}(\Theta_t) \right\rangle$$

$$= -\left\| \frac{1}{2n-1} \left( \sum_{i=1}^n x_i (y_i - \mathbb{E}_{p_{\Theta_t}(\cdot|x_i)}[y])^\top + \sum_{i=2}^n y_i (x_i - \mathbb{E}_{p_{\Theta_t}(\cdot|y_i)}[x])^\top \right) \right\|^2$$

$$= -\left\| \frac{1}{2n-1} \left( \sum_{i=1}^n \sum_{j=1}^m \alpha_{i,j}^{(t)} v_i v_j^\top + \sum_{i=n+2}^{2n} \sum_{j=1}^m \beta_{i,j}^{(t)} v_i v_j^\top \right) \right\|^2.$$

Similarly, we have

$$
\frac{dl_i(\Theta_t)}{dt}
$$

$$
= \left\langle \nabla l_i(\Theta_t), \frac{d\Theta_t}{dt} \right\rangle = - \left\langle \nabla l_i(\Theta_t), \nabla \mathcal{L}(\Theta_t) \right\rangle
$$

$$
= - \left\langle x_i(y_i - \mathbb{E}_{p_{\Theta_t}(\cdot|x_i)}[y])^\top, \frac{1}{2n-1} \left( \sum_{i=1}^{n} x_i(y_i - \mathbb{E}_{p_{\Theta_t}(\cdot|x_i)}[y])^\top + \sum_{i=2}^{n} y_i(x_i - \mathbb{E}_{p_{\Theta_t}(\cdot|y_i)}[x])^\top \right) \right\rangle
$$

$$
= - \left\langle \sum_{j=1}^{m} \alpha_{i,j} v_i v_j^\top, \frac{1}{2n-1} \left( \sum_{i=1}^{n}\sum_{j=1}^{m} \alpha_{i,j}^{(t)} v_i v_j^\top + \sum_{i=n+2}^{2n}\sum_{j=1}^{m} \beta_{i,j}^{(t)} v_i v_j^\top \right) \right\rangle,
$$

and

$$
\frac{dl_i^{\mathtt{rev}}(\Theta_t)}{dt}
$$

$$
= \left\langle \nabla l_i^{\mathtt{rev}}(\Theta_t), \frac{d\Theta_t}{dt} \right\rangle = - \left\langle \nabla l_i^{\mathtt{rev}}(\Theta_t), \nabla \mathcal{L}(\Theta_t) \right\rangle
$$

$$
= - \left\langle y_i(x_i - \mathbb{E}_{p_{\Theta_t}(\cdot|y_i)}[x])^\top, \frac{1}{2n-1} \left( \sum_{i=1}^{n} x_i(y_i - \mathbb{E}_{p_{\Theta_t}(\cdot|x_i)}[y])^\top + \sum_{i=2}^{n} y_i(x_i - \mathbb{E}_{p_{\Theta_t}(\cdot|y_i)}[x])^\top \right) \right\rangle
$$

$$
= - \left\langle \sum_{j=1}^{m} \beta_{i+n,j}^{(t)} v_{i+n} v_j^\top, \frac{1}{2n-1} \left( \sum_{i=1}^{n}\sum_{j=1}^{m} \alpha_{i,j}^{(t)} v_i v_j^\top + \sum_{i=n+2}^{2n}\sum_{j=1}^{m} \beta_{i,j}^{(t)} v_i v_j^\top \right) \right\rangle.
$$

Applying Lemma 7, with probability at least $1 - \delta/2$, for any $t \geq 0$ we have

$$
\left| \frac{d\mathcal{L}(\Theta_t)}{dt} + \frac{1}{(2n-1)^2} \left( \sum_{i=1}^{n}\sum_{j=1}^{m} (\alpha_{i,j}^{(t)})^2 + \sum_{i=n+2}^{2n}\sum_{j=1}^{m} (\beta_{i,j}^{(t)})^2 \right) \right|
$$
$$
\leq \epsilon \cdot \frac{1}{(2n-1)^2} \left( \sum_{i=1}^{n}\sum_{j=1}^{m} (\alpha_{i,j}^{(t)})^2 + \sum_{i=n+2}^{2n}\sum_{j=1}^{m} (\beta_{i,j}^{(t)})^2 \right),
$$
(5)

and

$$
\left| \frac{dl_i(\Theta_t)}{dt} + \frac{1}{2n-1} \sum_{j=1}^{m} (\alpha_{i,j}^{(t)})^2 \right|
$$
$$
\leq \epsilon \cdot \frac{1}{2n-1} \left( \sum_{j=1}^{m} (\alpha_{i,j}^{(t)})^2 \right)^{1/2} \left( \sum_{i=1}^{n}\sum_{j=1}^{m} (\alpha_{i,j}^{(t)})^2 + \sum_{i=n+2}^{2n}\sum_{j=1}^{m} (\beta_{i,j}^{(t)})^2 \right)^{1/2},
$$
(6)

$$
\left| \frac{dl_i^{\mathtt{rev}}(\Theta_t)}{dt} + \frac{1}{2n-1} \sum_{j=1}^{m} (\beta_{i+n,j}^{(t)})^2 \right|
$$
$$
\leq \epsilon \cdot \frac{1}{2n-1} \left( \sum_{j=1}^{m} (\beta_{i+n,j}^{(t)})^2 \right)^{1/2} \left( \sum_{i=1}^{n}\sum_{j=1}^{m} (\alpha_{i,j}^{(t)})^2 + \sum_{i=n+2}^{2n}\sum_{j=1}^{m} (\beta_{i,j}^{(t)})^2 \right)^{1/2}.
$$

Furthermore, Lemma 6 implies that with probability at least $1 - \delta/2$

$$
\frac{1}{2m} < p_{\Theta_0}(y_i|x_i), p_{\Theta_0}(x_i|y_i) < \frac{2}{m}.
$$
(7)

The following arguments are based on the event that the above inequalities hold.

We first show that

$$\inf\left\{p_{\Theta_t}(x_i|y_i), p_{\Theta_t}(y_i|x_i) : t \geq 0, i \in [n]\right\} > \frac{1}{2m}.$$

We prove this by contradiction. Let

$$\tau = \inf\left\{t \geq 0 : \exists i \in [n], \text{s.t.} \min\{p_{\Theta_t}(y_i|x_i), p_{\Theta_t}(x_i|y_i)\} \leq \frac{1}{2m}\right\}.$$

By Eq. (7), it is obvious that $\tau > 0$. Assume without loss of generality that $p_{\Theta_\tau}(y_1|x_1) \leq \frac{1}{2m}$. It follows that there exists $\delta > 0$ such that $p_{\Theta_t}(y_1|x_1)$ is a decreasing function in $(\tau - \delta, \tau)$ and $p_{\Theta_t}(y_1|x_1) = \min\{p_{\Theta_t}(x_i|y_i), p_{\Theta_t}(y_i|x_i) : i \in [n]\}$ for any $t \in (\tau - \delta, \tau)$. It follows that for $t \in (\tau - \delta, \tau)$,

$$
\begin{aligned}
\frac{dl_1(\Theta_t)}{dt} &\leq \frac{1}{2n-1}\left(-\sum_{j=1}^{m}(\alpha_{1,j}^{(t)})^2 + \epsilon \cdot \left(\sum_{j=1}^{m}(\alpha_{1,j}^{(t)})^2\right)^{1/2}\left(\sum_{i=1}^{n}\sum_{j=1}^{m}(\alpha_{i,j}^{(t)})^2 + \sum_{i=n+2}^{2n}\sum_{j=1}^{m}(\beta_{i,j}^{(t)})^2\right)^{1/2}\right) \\
&\leq \frac{1}{2n-1}\left(-\sum_{j=1}^{m}(\alpha_{1,j}^{(t)})^2 + \epsilon \cdot \left(\sum_{j=1}^{m}(\alpha_{1,j}^{(t)})^2\right)^{1/2}\left(2\sum_{i=1}^{n}(\alpha_{i,i+n}^{(t)})^2 + 2\sum_{i=n+2}^{2n}(\beta_{i,i-n}^{(t)})^2\right)^{1/2}\right) \\
&\leq \frac{1}{2n-1}\left(\sum_{j=1}^{m}(\alpha_{1,j}^{(t)})^2\right)^{1/2}\cdot\left(-\left(\sum_{j=1}^{m}(\alpha_{1,j}^{(t)})^2\right)^{1/2} + \epsilon \cdot \left(4n(\alpha_{1,n+1}^{(t)})^2\right)^{1/2}\right) \\
&\leq 0
\end{aligned}
$$

where the first inequality is from Eq. (6); the second inequality is due to $\sum_{j\neq i+n}|\alpha_{i,j}^{(t)}| = \alpha_{i,i+n}^{(t)} = 1 - p_{\Theta_t}(y_i|x_i), \sum_{j\neq i}|\beta_{i+n,j}^{(t)}| = \beta_{i+n,i}^{(t)} = 1 - p_{\Theta_t}(x_i|y_i)$ for all $i \in [n]$; the third inequality is because $p_{\Theta_t}(y_1|x_1) = \min\{p_{\Theta_t}(x_i|y_i), p_{\Theta_t}(y_i|x_i) : i \in [n]\}$. However, $p_{\Theta_t}(y_1|x_1)$ is a decreasing function in $(\tau - \delta, \tau)$, a contradiction. Therefore, we conclude that

$$\inf\left\{p_{\Theta_t}(x_i|y_i), p_{\Theta_t}(y_i|x_i) : t \geq 0, i \in [n]\right\} > \frac{1}{2m}.$$

Now we show

$$\mathcal{L}(\Theta_t) \leq \frac{1}{\frac{t}{8(2n-1)\log^2(2m)} + \frac{1}{\mathcal{L}(\Theta_0)}}, \quad \forall t \geq 0.$$

By Eq. (5),

$$
\begin{aligned}
\frac{d\mathcal{L}(\Theta_t)}{dt} &\leq -\frac{1-\epsilon}{(2n-1)^2}\left(\sum_{i=1}^{n}\sum_{j=1}^{m}(\alpha_{i,j}^{(t)})^2 + \sum_{i=n+2}^{2n}\sum_{j=1}^{m}(\beta_{i,j}^{(t)})^2\right) \\
&\leq -\frac{1-\epsilon}{(2n-1)^2}\left(\sum_{i=1}^{n}(1 - p_{\Theta_t}(y_i|x_i))^2 + \sum_{i=2}^{n}(1 - p_{\Theta_t}(x_i|y_i))^2\right) \\
&\leq -\frac{1-\epsilon}{(2n-1)^3}\left(\sum_{i=1}^{n}(1 - p_{\Theta_t}(y_i|x_i)) + \sum_{i=2}^{n}(1 - p_{\Theta_t}(x_i|y_i))\right)^2 \\
&\leq -\frac{1-\epsilon}{8(2n-1)\log^2(2m)}\mathcal{L}(\Theta_t)^2 \\
&\leq -\frac{1}{8(2n-1)\log^2(2m)}\mathcal{L}(\Theta_t)^2,
\end{aligned}
$$

where the second inequality is due to $\sum_{j\neq i+n}|\alpha_{i,j}^{(t)}| = \alpha_{i,i+n}^{(t)} = 1 - p_{\Theta_t}(y_i|x_i), \sum_{j\neq i}|\beta_{i+n,j}^{(t)}| = \beta_{i+n,i}^{(t)} = 1 - p_{\Theta_t}(x_i|y_i)$ for all $i \in [n]$; the third inequality applies Cauchy-Schwarz inequality;

the last inequality uses the fact that $p_{\Theta_t}(x_i|y_i), p_{\Theta_t}(y_i|x_i) > \frac{1}{2m}$ for any $t \geq 0, i \in [n]$ and the inequality $1 - x \geq \frac{\log x}{2 \log(1/(2m))}$ for $x \in (\frac{1}{2m}, 1)$.

By Lemma 10, we conclude that $\forall t \geq 0$,

$$\mathcal{L}(\Theta_t) \leq \frac{1}{\frac{t}{8(2n-1)\log^2(2m)} + \frac{1}{\mathcal{L}(\Theta_0)}},$$

which completes the proof. $\qquad\square$

**Lemma 4** (Dynamics of the reversal loss). *Let* $x_1, \ldots, x_n, y_1, \ldots, y_n, \mathcal{L}^{rev}(\Theta)$, *and* $\Theta_t$ $(t \geq 0)$ *be defined as in Theorem 5. When* $\sigma \leq \frac{1}{100 \ln(16n^2/\delta)}$ *and* $d \geq 400n^2 m^2 \log(8m^2 n^2/\delta)/\epsilon^2$, *with probability at least* $1 - \delta$,

$$\mathcal{L}^{rev}(\Theta_t) \geq \mathcal{L}^{rev}(\Theta_0) \cdot \left(1 + \frac{\mathcal{L}(\Theta_0) \cdot t}{8(2n-1)\log^2(2m)}\right)^{-8\epsilon(2n-1)\log^2(2m)}.$$

*Proof.* Similar to Lemma 3, we assume $x_1, \ldots, x_n = v_1, \ldots, v_n$ and $y_1, \ldots, y_n = v_{n+1}, \ldots v_{2n}$ WLOG. Let $\alpha_{i,j}^{(t)} = -p_{\Theta_t}(v_j|v_i) + \delta_{i,j-n}, \beta_{i,j}^{(t)} = -p_{\Theta_t}(v_j|v_i) + \delta_{i-n,j}$. By Lemma 5,

$$\frac{d\mathcal{L}^{\texttt{rev}}(\Theta_t)}{dt}$$

$$= \left\langle \nabla\mathcal{L}^{\texttt{rev}}(\Theta_t), \frac{d\Theta_t}{dt} \right\rangle$$

$$= -\left\langle y_1(x_1 - \mathbb{E}_{p_{\Theta_t}(\cdot|y_1)}[x])^\top, \frac{1}{2n-1}\left(\sum_{i=1}^n x_i(y_i - \mathbb{E}_{p_{\Theta_t}(\cdot|x_i)}[y])^\top + \sum_{i=2}^n y_i(x_i - \mathbb{E}_{p_{\Theta_t}(\cdot|y_i)}[x])^\top\right)\right\rangle$$

$$= -\left\langle \sum_{j=1}^m \beta_{n+1,j}^{(t)} v_{n+1} v_j^\top, \frac{1}{2n-1}\left(\sum_{i=1}^n \sum_{j=1}^m \alpha_{i,j}^{(t)} v_i v_j^\top + \sum_{i=n+2}^{2n} \sum_{j=1}^m \beta_{i,j}^{(t)} v_i v_j^\top\right)\right\rangle.$$

Applying Lemma 7, with probability at least $1 - \delta/2$, for any $t \geq 0$ we have

$$\left|\frac{d\mathcal{L}^{\texttt{rev}}(\Theta_t)}{dt}\right| \leq \epsilon \cdot \frac{1}{2n-1}\left(\sum_{j=1}^m (\beta_{n+1,j}^{(t)})^2\right)^{1/2}\left(\sum_{i=1}^n \sum_{j=1}^m (\alpha_{i,j}^{(t)})^2 + \sum_{i=n+2}^{2n} \sum_{j=1}^m (\beta_{i,j}^{(t)})^2\right)^{1/2}, \quad (8)$$

and

$$\left|\frac{d\mathcal{L}(\Theta_t)}{dt} + \frac{1}{(2n-1)^2}\left(\sum_{i=1}^n \sum_{j=1}^m (\alpha_{i,j}^{(t)})^2 + \sum_{i=n+2}^{2n} \sum_{j=1}^m (\beta_{i,j}^{(t)})^2\right)\right|$$

$$\leq \epsilon \cdot \frac{1}{(2n-1)^2}\left(\sum_{i=1}^n \sum_{j=1}^m (\alpha_{i,j}^{(t)})^2 + \sum_{i=n+2}^{2n} \sum_{j=1}^m (\beta_{i,j}^{(t)})^2\right),$$

as well as

$$\left|\frac{dl_i(\Theta_t)}{dt} + \frac{1}{2n-1}\sum_{j=1}^m (\alpha_{i,j}^{(t)})^2\right|$$

$$\leq \epsilon \cdot \frac{1}{2n-1}\left(\sum_{j=1}^m (\alpha_{i,j}^{(t)})^2\right)^{1/2}\left(\sum_{i=1}^n \sum_{j=1}^m (\alpha_{i,j}^{(t)})^2 + \sum_{i=n+2}^{2n} \sum_{j=1}^m (\beta_{i,j}^{(t)})^2\right)^{1/2},$$

$$\left|\frac{dl_i^{\texttt{rev}}(\Theta_t)}{dt} + \frac{1}{2n-1}\sum_{j=1}^m (\beta_{i+n,j}^{(t)})^2\right|$$

$$\leq \epsilon \cdot \frac{1}{2n-1}\left(\sum_{j=1}^m (\beta_{i+n,j}^{(t)})^2\right)^{1/2}\left(\sum_{i=1}^n \sum_{j=1}^m (\alpha_{i,j}^{(t)})^2 + \sum_{i=n+2}^{2n} \sum_{j=1}^m (\beta_{i,j}^{(t)})^2\right)^{1/2}.$$

Furthermore, Lemma 6 implies that with probability at least $1 - \delta/2$,

$$\frac{1}{2m} \le p_{\Theta_0}(y_i|x_i), p_{\Theta_0}(x_i|y_i) \le \frac{2}{m}. \tag{9}$$

The following arguments are based on the event that the above inequalities hold.

By Eq. (8),

$$\frac{d\mathcal{L}^{\mathtt{rev}}(\Theta_t)}{dt} \ge -\epsilon \cdot \frac{1}{2n-1} \underbrace{\left( \sum_{j=1}^{m} (\beta_{n+1,j}^{(t)})^2 \right)^{1/2}}_{A_t} \underbrace{\left( \sum_{i=1}^{n} \sum_{j=1}^{m} (\alpha_{i,j}^{(t)})^2 + \sum_{i=n+2}^{2n} \sum_{j=1}^{m} (\beta_{i,j}^{(t)})^2 \right)^{1/2}}_{B_t}.$$

Notice that

$$B_t \le 2 \sum_{i=1}^{n} (1 - p_{\Theta_t}(y_i|x_i))^2 + 2 \sum_{i=2}^{n} (1 - p_{\Theta_t}(x_i|y_i))^2$$

$$\le 2 \left( \sum_{i=1}^{n} (1 - p_{\Theta_t}(y_i|x_i)) + \sum_{i=2}^{n} (1 - p_{\Theta_t}(x_i|y_i)) \right)^2$$

$$\le 2(2n-1)^2 \mathcal{L}(\Theta_t)^2$$

$$\le 2(2n-1)^2 \left( \frac{1}{\frac{t}{8(2n-1)\log^2(2m)} + \frac{1}{\mathcal{L}(\Theta_0)}} \right)^2$$

where the first inequality uses $\sum_{j\ne i+n} |\alpha_{i,j}^{(t)}| = \alpha_{i,i+n}^{(t)} = 1 - p_{\Theta_t}(y_i|x_i), \sum_{j\ne i} |\beta_{i+n,j}^{(t)}| = \beta_{i+n,i}^{(t)} = 1 - p_{\Theta_t}(x_i|y_i)$ for all $i \in [n]$; the third inequality uses the fact that $1 - x \le -\log x$; the last inequality applies Lemma 3.

Similarly,

$$A_t \le 2(1 - p_{\Theta_t}(x_1|y_1))^2 \le 2\mathcal{L}^{\mathtt{rev}}(\Theta_t)^2.$$

Combining, we have

$$\frac{d\mathcal{L}^{\mathtt{rev}}(\Theta_t)}{dt} \ge -\epsilon \cdot \frac{1}{2n-1} \cdot 2\mathcal{L}^{\mathtt{rev}}(\Theta_t) \cdot (2n-1) \cdot \frac{1}{\frac{t}{8(2n-1)\log^2(2m)} + \frac{1}{\mathcal{L}(\Theta_0)}}$$

$$\ge -8\epsilon(2n-1)\log^2(2m) \cdot \mathcal{L}^{\mathtt{rev}}(\Theta_t) \cdot \frac{1}{t + \frac{8(2n-1)\log^2(2m)}{\mathcal{L}(\Theta_0)}}.$$

By Lemma 10, we conclude that $\forall t \ge 0$,

$$\mathcal{L}^{\mathtt{rev}}(\Theta_t) \ge \mathcal{L}^{\mathtt{rev}}(\Theta_0) \cdot \left( 1 + \frac{\mathcal{L}(\Theta_0) \cdot t}{8(2n-1)\log^2(2m)} \right)^{-8\epsilon(2n-1)\log^2(2m)},$$

This completes the proof. $\qquad\square$

**Lemma 5** (Gradient of the loss function). *Define*

$$\mathcal{L}(\Theta) = \frac{1}{2n-1} \left( \sum_{i=1}^{n} -\log p_{\Theta}(y_i|x_i) + \sum_{i=2}^{n} -\log p_{\Theta}(x_i|y_i) \right),$$

$$\mathcal{L}^{rev}(\Theta) = -\log p_{\Theta}(x_1|y_1).$$

*Then we have*

$$\nabla\mathcal{L}(\Theta) = -\frac{1}{2n-1} \left( \sum_{i=1}^{n} x_i (y_i - \mathbb{E}_{p_{\Theta}(\cdot|x_i)}[y])^\top + \sum_{i=2}^{n} y_i (x_i - \mathbb{E}_{p_{\Theta}(\cdot|y_i)}[x])^\top \right),$$

$$\nabla\mathcal{L}^{rev}(\Theta) = -y_1 (x_1 - \mathbb{E}_{p_{\Theta}(\cdot|y_1)}[x])^\top.$$

*Proof.* We have

$$\nabla(-\log p_\Theta(y|x))$$

$$= -\frac{\nabla p_\Theta(y|x)}{p_\Theta(y|x)}$$

$$= -\frac{1}{p_\Theta(y|x)} \cdot \frac{xy^\top \exp(x^\top \Theta y)\left(\sum_{y\in\mathcal{V}} \exp(x^\top \Theta y)\right) - \exp(x^\top \Theta y)\left(\sum_{y\in\mathcal{V}} xy^\top \exp(x^\top \Theta y)\right)}{\left(\sum_{y\in\mathcal{V}} \exp(x^\top \Theta y)\right)^2}$$

$$= -\frac{1}{p_\Theta(y|x)}\left(p_\Theta(y|x)xy^\top - p_\Theta(y|x)\sum_{y\in\mathcal{V}} p_\Theta(y|x)xy^\top\right)$$

$$= -x\left(y - \sum_{y\in\mathcal{V}} p_\Theta(y|x)y\right)^\top$$

$$= -x\left(y - \mathbb{E}_{y\sim p_\Theta(\cdot|x)}[y]\right)^\top.$$

The statements follow immediately. $\qquad\square$

## B.2 Initialization

**Lemma 6** (Initial distributions are all close to uniform). *Fix any $\delta \in (0,1)$. Let $x_1, x_2, \ldots, x_n \overset{i.i.d.}{\sim} \mathcal{N}_d(\mathbf{0}_d, \frac{1}{d}I_d)$. Let $\Theta \in \mathbb{R}^{d\times d}$, where each $\Theta_{ij} \overset{i.i.d.}{\sim} \mathcal{N}(0, \sigma^2)$ independent of $x_1, \ldots, x_n$. For any $i, j \in [n]$, define*

$$p_\Theta(x_j|x_i) = \frac{\exp(l_\Theta(x_j|x_i))}{\sum_{k=1}^n \exp(l_\Theta(x_k|x_i))}, \quad \text{where} \quad l_\Theta(x_j|x_i) = x_i^\top \Theta x_j.$$

*Then when $\sigma^2 \le \frac{1}{100 \ln(4n^2/\delta)}$ and $d \ge 400 \log(1/(2\delta n^2))/\epsilon^2$, with probability at least $1 - \delta$,*

$$|p_\Theta(x_j|x_i) - 1/n| \le \frac{1}{2n}, \ \forall i, j \in [n].$$

*Proof.* Let $v = 0.2$. By Lemma 8, with probability at least $1 - \delta$, we have

$$|\langle x_i, x_j \rangle - \delta_{ij}| \le v, \ \forall i, j \in [n].$$

Conditioned on the above high-probability event, we can further obtain that for any $j \in [n]$

$$p_\Theta(x_j|x_i) = \frac{\exp(l_\Theta(x_j|x_i))}{\sum_{k=1}^n \exp(l_\Theta(x_k|x_i))} \le \frac{\exp(v)}{\sum_{k=1}^n \exp(-v)} = \frac{\exp(2v)}{n},$$

and

$$p_\Theta(x_j|x_i) = \frac{\exp(l_\Theta(x_j|x_i))}{\sum_{k=1}^n \exp(l_\Theta(x_k|x_i))} \ge \frac{\exp(-v)}{\sum_{k=1}^n \exp(v)} = \frac{\exp(-2v)}{n},$$

It follows that

$$\frac{1}{2n} < p_\Theta(x_j|x_i) < \frac{3}{2n} \implies |p_\Theta(x_j|x_i) - 1/n| < \frac{1}{2n}.$$

which completes the proof. $\qquad\square$

## B.3 Subspace embedding

**Lemma 7** ($\ell_1$-subspace embedding of Gaussian second-order tensors). *Let $z_1, \ldots, z_n$ be independently sampled from $\mathcal{N}_d(\mathbf{0}_d, \frac{1}{d}I_d)$. Let $\mathcal{I}_1, \mathcal{I}_2 \subseteq [n] \times [n]$ be two index sets. Let $\mathcal{I}_0 = \mathcal{I}_1 \cap \mathcal{I}_2$. If $d \ge 64 \log(2n^2/\delta)/\epsilon^2$, then with probability at least $1 - \delta$,*

$$\left|\left\langle \sum_{(i,j)\in\mathcal{I}_1} \alpha_{i,j} z_i z_j^\top, \sum_{(i,j)\in\mathcal{I}_2} \beta_{i,j} z_i z_j^\top \right\rangle - \sum_{(i,j)\in\mathcal{I}_0} \alpha_{i,j}\beta_{i,j}\right| \le \epsilon \cdot \left(\sum_{(i,j)\in\mathcal{I}_1} |\alpha_{i,j}|\right)\left(\sum_{(i,j)\in\mathcal{I}_2} |\beta_{i,j}|\right)$$

*holds for any $\alpha_{i,j}, \beta_{i,j}$. Furthermore, if $d \geq 64k^2 \log(2n^2/\delta)/\epsilon^2$, then with probability at least $1 - \delta$,*

$$\left| \left\langle \sum_{(i,j) \in \mathcal{I}_1} \alpha_{i,j} z_i z_j^\top, \sum_{(i,j) \in \mathcal{I}_2} \beta_{i,j} z_i z_j^\top \right\rangle - \sum_{(i,j) \in \mathcal{I}_0} \alpha_{i,j} \beta_{i,j} \right| \leq \epsilon \cdot \left( \sum_{(i,j) \in \mathcal{I}_1} \alpha_{i,j}^2 \right)^{1/2} \left( \sum_{(i,j) \in \mathcal{I}_2} \beta_{i,j}^2 \right)^{1/2}$$

*holds for any $\alpha_{i,j}, \beta_{i,j}$ such that $\|\alpha\|_0 \leq k, \|\beta\|_0 \leq k$.*

*Proof.* Using Lemma 8 and Cauchy-Schwarz inequality, we have

$$\left| \left\langle \sum_{(i,j) \in \mathcal{I}_1} \alpha_{i,j} z_i z_j^\top, \sum_{(i,j) \in \mathcal{I}_2} \beta_{i,j} z_i z_j^\top \right\rangle - \sum_{(i,j) \in \mathcal{I}_0} \alpha_{i,j} \beta_{i,j} \right|$$

$$\leq \left| \sum_{(i,j) \in \mathcal{I}_0} \alpha_{i,j} \beta_{i,j} (\|z_i\|^2 \|z_j\|^2 - 1) + \sum_{(i,j) \in \mathcal{I}_1, (k,l) \in \mathcal{I}_2, (i,j) \neq (k,l)} \alpha_{i,j} \beta_{k,l} z_i^\top z_k z_j^\top z_l \right|$$

$$\leq \sqrt{\frac{64 \log(2n^2/\delta)}{d}} \cdot \left( \sum_{(i,j) \in \mathcal{I}} |\alpha_{i,j} \beta_{i,j}| + \sum_{(i,j) \in \mathcal{I}_1, (k,l) \in \mathcal{I}_2, (i,j) \neq (k,l)} |\alpha_{i,j} \beta_{k,l}| \right)$$

$$= \sqrt{\frac{64 \log(2n^2/\delta)}{d}} \cdot \left( \sum_{(i,j) \in \mathcal{I}_1} |\alpha_{i,j}| \right) \cdot \left( \sum_{(i,j) \in \mathcal{I}_2} |\beta_{i,j}| \right)$$

$$\leq \sqrt{\frac{64 k^2 \log(2n^2/\delta)}{d}} \cdot \left( \sum_{(i,j) \in \mathcal{I}_2} \alpha_{i,j}^2 \right)^{1/2} \left( \sum_{(i,j) \in \mathcal{I}_2} \beta_{i,j}^2 \right)^{1/2}.$$

The statements follow directly from plugging in suitable values of $d$. $\square$

**Lemma 8** (Almost orthonormal). *Let $x_1, x_2, \ldots, x_n \overset{i.i.d.}{\sim} \mathcal{N}_d(\mathbf{0}_d, \frac{1}{d} I_d)$. For any $\epsilon, \delta \in (0, 1)$, when $d \geq 16 \log(2n^2/\delta)/\epsilon^2$, it holds that with probability at least $1 - \delta$,*

$$|\langle x_i, x_j \rangle - \delta_{ij}| \leq \epsilon, \ \forall i, j \in [n].$$

*Proof.* Fix $i \neq j \in [n]$. Notice

$$|\langle x_i, x_j \rangle| = |\langle x_i + x_j, x_i + x_j \rangle - \langle x_i - x_j, x_i - x_j \rangle|/4$$
$$\leq (|\langle x_i + x_j, x_i + x_j \rangle - 1| + |\langle x_i - x_j, x_i - x_j \rangle - 1|)/4.$$

Using Lemma 9, we have that with probability at least $1 - \delta/n^2$,

$$|\langle x_i, x_j \rangle| \leq \epsilon.$$

Furthermore, fix $i \in [n]$, with probability at least $1 - \delta/n^2$,

$$|\langle x_i, x_i \rangle - 1| \leq \epsilon.$$

The statement then follows from union bound over $i, j \in [n]$. $\square$

**Lemma 9** (Almost normal for a fixed vector). *For a $d$-dimensional random vector $x \sim \mathcal{N}_d(\mathbf{0}_d, \frac{1}{d} I_d)$ and any $v \in (0, 1/2)$,*

$$\mathbb{P}\left( |\langle x, x \rangle - 1| \geq v \right) \leq 2 e^{-v^2 d/16}.$$

*In particular, when $d \geq 16 \log(1/(2\delta))/v^2$, we have*

$$\mathbb{P}\left( |\langle x, x \rangle - 1| \geq v \right) \leq \delta$$

*Proof.* Let $x = (x_1, \ldots, x_d)$. By Lemma 11 (letting $x = v^2 d/16$),

$$\mathbb{P}\left( |\langle x, x \rangle - 1| \geq v \right) \leq \mathbb{P}\left( \left| \sum_{i=1}^d (\sqrt{d} \cdot x_i)^2 - d \right| \geq vd \right)$$

$$\leq 2 e^{-v^2 d/16}.$$

The second inequality follows from simple arithmetics. $\square$

### B.4 Useful results

**Lemma 10** (ODE bound). *Let $c_1, c_2, c_3 > 0$. Suppose the function $f_1, f_2 : \mathbb{R}_+ \to \mathbb{R}$ satisfies $f_1(0) > 0, f_2(0) > 0$ and*

$$\frac{df_1(t)}{dt} \leq -c_1 \cdot f_1(t)^2,$$

$$\frac{df_2(t)}{dt} \geq -c_2 \cdot f_2(t) \cdot \frac{1}{t + c_3}.$$

*Then*

$$f_1(t) \leq \frac{1}{c_1 t + \frac{1}{f_1(0)}},$$

$$f_2(t) \geq f_2(0) \cdot \left(1 + \frac{t}{c_3}\right)^{-c_2}.$$

*Proof.* The conditions imply that

$$\frac{df_1^{-1}(t)}{dt} = -\frac{1}{f_1^2(t)} \cdot \frac{df_1(t)}{dt} \geq c_1,$$

$$\frac{d\log f_2(t)}{dt} = \frac{1}{f_2(t)} \cdot \frac{df_2(t)}{dt} \geq -c_2/(t + c_3).$$

It follows that

$$f_1^{-1}(t) \geq c_1 t + f_1^{-1}(0),$$

$$\log f_2(t) \geq -c_2 \log(1 + t/c_3) + \log f_2(0).$$

Rearranging the above inequalities, one can obtain the desired results. $\qquad\square$

**Lemma 11** ($\chi^2$-concentration bound, Lemma 1 of [64]). *Let $g_1, \ldots, g_t$ be i.i.d. $\mathcal{N}(0, 1)$ random variables. Then for any $x \geq 0$,*

$$\Pr\left[\sum_{i=1}^t g_i^2 \geq t + 2\sqrt{tx} + 2x\right] \leq \exp(-x),$$

*and*

$$\Pr\left[\sum_{i=1}^t g_i^2 \leq t - 2\sqrt{tx}\right] \leq \exp(-x).$$

## C    Missing Proofs of Section 4

In this section, we show missing proofs in Section 4.

### C.1    Proofs of Section 4.1

#### C.1.1    Proofs of Proposition 4.1 and Proposition 4.2

We first show the proofs of Proposition 4.1 and Proposition 4.2, respectively.

*Proof of Proposition 4.1.* Actually, for any three tokens $x_1, x_2, x_3$, it holds that

$$p_{\theta(0)}(x_3 | x_1, x_2) = \frac{\exp\left(\boldsymbol{x}_3^\top Y(0)^\top \mathrm{LN}(X^\top \boldsymbol{b}_2)\right)}{\sum_{x' \in [M]} \exp\left(\boldsymbol{x}'^\top Y(0)^\top \mathrm{LN}(X^\top \boldsymbol{b}_2)\right)} = \frac{\exp(0)}{\sum_{x' \in [M]} \exp(0)} = 1/M,$$

since $Y(0) = \mathbf{0}$. $\qquad\square$

*Proof of Proposition 4.2.* Note that the input sequence length $T = 2$. By (1),

$$p_{\theta(t)}(x|x_1, x_2) = \frac{\exp\left(\boldsymbol{x}^\top Y(t)^\top \mathrm{LN}(X^\top \boldsymbol{b}_2)\right)}{\sum_{x' \in [M]} \exp\left(\boldsymbol{x}'^\top Y(t)^\top \mathrm{LN}(X^\top \boldsymbol{b}_2)\right)}$$

where $\boldsymbol{b}_2 = [b_{12}]$ and $b_{12} = 1$. Also, $X = [\boldsymbol{x}_1]^\top$ is a one-hot row vector. Therefore, $\mathrm{LN}(X^\top \boldsymbol{b}_2) = \mathrm{LN}(\boldsymbol{x}_1 b_{12}) = \mathrm{LN}(\boldsymbol{x}_1) = \boldsymbol{x}_1$, and thus

$$p_{\theta(t)}(x|x_1, x_2) = \frac{\exp\left(\boldsymbol{x}^\top Y(t)^\top \boldsymbol{x}_1\right)}{\sum_{x' \in [M]} \exp\left(\boldsymbol{x}'^\top Y(t)^\top \boldsymbol{x}_1\right)} = \frac{\exp\left(Y(t)_{x_1, x}\right)}{\sum_{x' \in [M]} \exp\left(Y(t)_{x_1, x'}\right)}$$

where $Y(t)_{i,j}$ is the entry of the matrix $Y(t)$ at row $i$ and column $j$.

$\square$

### C.1.2  Proof of Lemma 2

*Proof of Lemma 2.* We first calculate the gradient of $Y$ when the current batch is a sequence $(x_1, x_2, x_3)$. Note that the input sequence length $T = 2$ and by the proof of Proposition 4.2, we have $\mathrm{LN}(X^\top \boldsymbol{b}_T) = \mathrm{LN}(X^\top \boldsymbol{b}_2) = \boldsymbol{x}_1$. By Lemma 1, we have

$$\dot{Y} = \eta_Y \mathrm{LN}(X^\top \boldsymbol{b}_T)(\boldsymbol{x}_{T+1} - \boldsymbol{\alpha})^\top = \eta_Y \boldsymbol{x}_1 (\boldsymbol{x}_3 - \boldsymbol{\alpha})^\top$$

where $\boldsymbol{\alpha} = [\alpha_1, \alpha_2, \ldots, \alpha_M]^\top \in \mathbb{R}^M$ with $\boldsymbol{\alpha} = \exp\left(Y_{x_1}^\top\right)/\mathbf{1}^\top \exp\left(Y_{x_1}^\top\right)$. Note that $\boldsymbol{x}_1 (\boldsymbol{x}_3 - \boldsymbol{\alpha})^\top$ is a matrix with only $x_1$-th row non-zero since $\boldsymbol{x}_1$ is one-hot. Therefore, the update of each row of $Y$ are independent and only $x_1$-th row of $Y$ gets updated at the current time step.

Now we consider any fixed $x_1 \in \mathcal{V}$. Let $t_{x_1, i}$ be the time step that the $x_1$-th row of Y gets updated (i.e., the first token of the training data is $x_1$ for the current batch) for the $i$-th time and let $t_{x_1, 0} = 0$ for notation convenience, then

$$Y(t_{x_1, i})_{x_1} = Y(t_{x_1, i-1})_{x_1} + \eta_Y (\boldsymbol{x}_3 - \boldsymbol{\alpha})^\top.$$

For convenience, we denote $\boldsymbol{y}(i) = Y(t_{x_1, i})_{x_1}^\top$, and thus

$$\boldsymbol{y}(i) = \boldsymbol{y}(i-1) + \eta_Y (\boldsymbol{x}_3 - \boldsymbol{\alpha}(i-1)) \tag{10}$$

where $\boldsymbol{y}(0) = \mathbf{0}$ and $\boldsymbol{\alpha}(i-1) = \exp(\boldsymbol{y}(i-1))/\mathbf{1}^\top \exp(\boldsymbol{y}(i-1))$. Note that for a fixed $x_1$, $x_3$ is also fixed by our construction of the dataset. By Lemma 5 of [14], we can obtain that

$$\boldsymbol{y}(i) = (M-1)h^*(i)\boldsymbol{\xi}_{x_3},$$

where $\boldsymbol{\xi}_{x_3} = \frac{M}{M-1}(\boldsymbol{x}_3 - \frac{1}{M}\mathbf{1})$ and $h^*(i)$ can be derived recursively as

$$h^*(i) = h^*(i-1) + \frac{\eta_Y}{(M-1) + \exp(Mh^*(i-1))}$$

with $h^*(0) = 0$. Combining Lemma 7 and 9 in [14], we have

$$h^*(i) \gtrsim \frac{1}{M}\ln(M\eta_Y i), \quad \forall i \gtrsim \frac{\ln M}{\eta_Y}. \tag{11}$$

Note that the update of each row of $Y$ are independent, the training set has size $N$, and the batch size is 1, we have

$$Y(t)_{x_1}^\top = (M-1)h^*\left(\lceil t/N \rceil\right)\boldsymbol{\xi}_{x_3}$$

where the training data at time step $t$ is $(x_1, x_2, x_3)$. Combining (11), we can obtain that

$$Y(t)_{x_1, x_3} \gtrsim (M-1) \cdot \frac{M}{M-1}(1 - \frac{1}{M}) \cdot \frac{1}{M}\ln\left(M\eta_Y \lceil t/N \rceil\right) \geq \ln\left(\frac{M\eta_Y t}{N}\right)$$

and

$$Y(t)_{x_1, x} \lesssim (M-1) \cdot \frac{M}{M-1}(-\frac{1}{M}) \cdot \frac{1}{M}\ln\left(M\eta_Y \lceil t/N \rceil\right) \leq -\frac{1}{M}\ln\left(\frac{M\eta_Y t}{N}\right), \quad \forall x \neq x_3.$$

On the other hand, for any sequence $(x_1, x_2, x_3)$ in the test set, since the $x_1$-th row of $Y$ has never been updated, we have

$$Y(t)_{x_1, x} = Y(0)_{x_1, x} = 0, \quad \forall x \in [M].$$

$\square$

### C.1.3 Proof of Theorem 3

*Proof.* We first consider training sequence $(x_1, x_2, x_3)$ at time $t$. By Proposition 4.2, we have

$$p_{\theta(t)}(x_3|x_1, x_2) = \frac{\exp\left(Y(t)_{x_1,x_3}\right)}{\sum_{x' \in [M]} \exp\left(Y(t)_{x_1,x'}\right)}$$

and by Lemma 2, we have

$$Y(t)_{x_1,x_3} \geq c\ln\left(\frac{M\eta_Y t}{N}\right), \quad \text{and} \quad Y(t)_{x_1,x} \leq -\frac{c}{M}\ln\left(\frac{M\eta_Y t}{N}\right), \quad \forall x \neq x_3$$

for some constant $c > 0$. Therefore,

$$p_{\theta(t)}(x_3|x_1, x_2) \geq \frac{\exp\left(c\ln\left(\frac{M\eta_Y t}{N}\right)\right)}{\exp\left(c\ln\left(\frac{M\eta_Y t}{N}\right)\right) + (M-1)\ln\left(-\frac{c}{M}\ln\left(\frac{M\eta_Y t}{N}\right)\right)}$$

$$\geq \frac{\left(\frac{M\eta_Y t}{N}\right)^c}{\left(\frac{M\eta_Y t}{N}\right)^c + (M-1)}$$

$$= 1 - \frac{M-1}{\left(\frac{M\eta_Y t}{N}\right)^c + (M-1)}$$

$$\geq 1 - \frac{M-1}{2\left(\frac{M\eta_Y t}{N}\right)^c}$$

where the last inequality holds since

$$\ln t \gtrsim \ln(NM/\eta_Y) \implies t \geq \frac{N(M-1)^{1/c}}{M\eta_Y} \implies \left(\frac{M\eta_Y t}{N}\right)^c \geq M-1.$$

Finally, for any sequence $(x_1, x_2, x_3) \in \mathcal{D}_{\text{test}}$, since $Y(t)_{x_1,x} = 0, \forall x \in [M]$ according to Lemma 2, we have

$$p_{\theta(t)}(x_3|x_1, x_2) = \frac{\exp\left(Y(t)_{x_1,x_3}\right)}{\sum_{x' \in [M]} \exp\left(Y(t)_{x_1,x'}\right)} = \frac{\exp(0)}{M \cdot \exp(0)} = 1/M,$$

which completes the proof. $\qquad\square$

### C.2 Additional details and proof of Section 4.2

**Datasets.** Let $N_{\text{train}} > 0$, $N_{\text{test}} > 0$ be two positive integers and let $N_{\text{total}} = N_{\text{train}} + N_{\text{test}}$. Let $\text{A}_i, \text{B}_i, \text{C}_i \in \mathcal{V}, \forall i \in [N_{\text{total}}]$ be $3N_{\text{total}}$ distinct tokens. Let $\rightarrow, \rightsquigarrow \in \mathcal{V} = [M]$ be two additional different tokens that represent "direct implication" and "indirect implication" respectively. Specifically, we have $\text{A}_i \rightarrow \text{B}_i$, $\text{B}_i \rightarrow \text{C}_i$ and $\text{A}_i \rightsquigarrow \text{C}_i$ for all $i \in [N_{\text{total}}]$. For notation convenience, we define the following two index sets

$$\mathcal{I}_{\text{train}} = \{1, 2, \ldots, N_{\text{train}}\}, \quad \mathcal{I}_{\text{test}} = \{N_{\text{train}} + 1, \ldots, N_{\text{total}}\}.$$

The training set $\mathcal{D}_{\text{train}}$ consists of all $\text{A}_i \rightarrow \text{B}_i$, $\text{B}_i \rightarrow \text{C}_i$ and $\text{A}_i \rightsquigarrow \text{C}_i$ for $i \in \mathcal{I}_{\text{train}}$. In addition, $\mathcal{D}_{\text{train}}$ contains $\text{A}_i \rightarrow \text{B}_i$ and $\text{B}_i \rightarrow \text{C}_i$ for $i \in \mathcal{I}_{\text{test}}$. For convenience, we let $N = |\mathcal{D}_{\text{train}}|$ to be the size of the training set. The test set $\mathcal{D}_{\text{test}}$ consists of $\text{A}_i \rightsquigarrow \text{C}_i$ for $i \in \mathcal{I}_{\text{test}}$. Under our construction of the dataset, the LLM will learn the relationship between $\text{A}_i, \text{B}_i$ and $\text{C}_i$ for $i \in \mathcal{I}_{\text{train}}$ in both direct and indirect implication, and learn the relationship between $\text{A}_i, \text{B}_i$ and $\text{C}_i$ for $i \in \mathcal{I}_{\text{test}}$ only in direct implication and will be tested for indirect implication.

Similar to the reversal curse in Section 4.1, we aim to prove through the training dynamics of one-layer transformers that the test probability remains negligible during training. In particular, we are interested in

$$p_\theta(x_3 = \text{C}_i|x_1 = \text{A}_i, x_2 = \rightsquigarrow), \quad i \in \mathcal{I}_{\text{test}}.$$

We also use $p_\theta(\text{B}_i|\text{A}_i \rightarrow)$, $p_\theta(\text{C}_i|\text{B}_i \rightarrow)$ and $p_\theta(\text{C}_i|\text{A}_i \rightsquigarrow)$ to more compactly represent $p_\theta(x_3 = \text{B}_i|x_1 = \text{A}_i, x_2 = \rightarrow)$, $p_\theta(x_3 = \text{C}_i|x_1 = \text{B}_i, x_2 = \rightarrow)$ and $p_\theta(x_3 = \text{C}_i|x_1 = \text{A}_i, x_2 = \rightsquigarrow)$, respectively.

The following theorem shows the importance of the chain-of-thought method:

**Theorem 7** (Importance of chain-of-thought, formal statement of Theorem 4). *Assume we run SGD with batch size 1, and assume $M \gg 100$ and $\frac{1}{M^{0.99}} \ll \eta_Y < 1$. Let $t \gtrsim \frac{N \ln M}{\eta_Y}$ denote the time step which also satisfies $\ln t \gtrsim \ln(NM/\eta_Y)$. For any test index $i \in \mathcal{I}_{test}$, we have*

$$p_{\theta(t)}(B_i|A_i \rightarrow) \geq 1 - \frac{M-1}{2\left(\frac{M\eta_Y t}{N}\right)^c}, \qquad p_{\theta(t)}(C_i|B_i \rightarrow) \geq 1 - \frac{M-1}{2\left(\frac{M\eta_Y t}{N}\right)^c}$$

*for some constant $c > 0$ and*

$$p_{\theta(t)}(C_i|A_i \rightsquigarrow) \leq \frac{1}{M}.$$

*Proof.* Recall that by Proposition 4.2, we have

$$p_{\theta(t)}(x_3|x_1, x_2) = \frac{\exp\left(Y(t)_{x_1,x_3}\right)}{\sum_{x' \in [M]} \exp\left(Y(t)_{x_1,x'}\right)}$$

and by Lemma 2, we have

$$Y(t)_{A_i,B_i} \geq c \ln\left(\frac{M\eta_Y t}{N}\right), \quad \text{and} \quad Y(t)_{A_i,x} \leq -\frac{c}{M} \ln\left(\frac{M\eta_Y t}{N}\right), \quad \forall x \neq B_i$$

for some constant $c > 0$. Therefore, using the same proof as Theorem 3, we have

$$p_{\theta(t)}(B_i|A_i \rightarrow) \geq 1 - \frac{M-1}{2\left(\frac{M\eta_Y t}{N}\right)^c}.$$

Additionally, according to the proof of Lemma 2, $Y(t)_{A_i,x}$ has the same value across all $x \neq B_i$, which implies

$$p_{\theta(t)}(C_i|A_i \rightsquigarrow) \leq \frac{1}{M}.$$

Similarly, applying Lemma 2 to $Y(t)_{B_i}$, we can obtain that

$$p_{\theta(t)}(C_i|B_i \rightarrow) \geq 1 - \frac{M-1}{2\left(\frac{M\eta_Y t}{N}\right)^c}.$$

$\square$

### C.3 Analysis of the reversal curse for the four-token sequences

In this section, we analyze the reversal curse where data points are four-token sentences "$AR_1R_2B$" or "$BR_1R_2A$". For each sentence, $A$ and $B$ are two distinct tokens that represent two entities, and $R_1$, $R_2$ are two special tokens jointly representing a relationship that is inverse to itself (e.g., $R_1R_2$ represents "is the friend of", then $AR_1R_2B$ means "A is the friend of B" and $BR_1R_2A$ means "B is the friend of A").

**Datasets.** Let $N_{\text{train}} > 0$ and $N_{\text{test}} > 0$ and denote $N_{\text{total}} = N_{\text{train}} + N_{\text{test}}$. Let $A_i, B_i \in \mathcal{V}, \forall i \in [N_{\text{total}}]$ be $K \triangleq 2N_{\text{total}}$ distinct tokens representing distinct entities. WLOG, we assume $A_i, B_i \in [K], \forall i \in [N_{\text{total}}]$. Let $R_1, R_2 \in \mathcal{V}$ be two additional different tokens that jointly represent a relationship that is inverse to itself. Specifically, we have $A_iR_1R_2B_i$ and $B_iR_1R_2A_i$ for all $i \in [N_{\text{total}}]$. For notation convenience, we define the following two index sets

$$\mathcal{I}_{\text{train}} = [N_{\text{train}}], \qquad \mathcal{I}_{\text{test}} = [N_{\text{total}}]\backslash\mathcal{I}_{\text{train}}.$$

The training set $\mathcal{D}_{\text{train}}$ consists of all $A_iR_1R_2B_i$ and $B_iR_1R_2A_i$ for $i \in \mathcal{I}_{\text{train}}$. In addition, $\mathcal{D}_{\text{train}}$ contains $A_iR_1R_2B_i$ for $i \in \mathcal{I}_{\text{test}}$. For convenience, we let $N = |\mathcal{D}_{\text{train}}|$ to be the size of the training set. The test set $\mathcal{D}_{\text{test}}$ consists of $B_iR_1R_2A_i$ for $i \in \mathcal{I}_{\text{test}}$. Under our construction of the dataset, the LLM will learn the relationship between $A_i$ and $B_i$ for $i \in \mathcal{I}_{\text{train}}$ in both directions to deduce that the relationship "$R_1R_2$" is reverse to itself, and learn the relationship between $A_i$ and $B_i$ for $i \in \mathcal{I}_{\text{test}}$ in one direction and will be tested for the other. WLOG, we assume for each training sequence $(x_1, R_1, R_2, x_4) \in \mathcal{D}_{\text{train}}$, $x_4 \in [N]$.

Now we assume that the learning rate $\eta_Y \gg \eta_Z$ and therefore can treat $X^\top \boldsymbol{b}_T$ as fixed when analyzing the dynamics of $Y$. For any sequence $(x_1, x_2 = \mathtt{R}_1, x_3 = \mathtt{R}_2, x_4 = n)$ in training or test dataset with $T = 3$, we define $\boldsymbol{f}_n = \mathrm{LN}(X^\top \boldsymbol{b}_T)$. Then the gradient of $Y$ in (3) becomes

$$\dot{Y} = \eta_Y \boldsymbol{f}_n (\boldsymbol{x}_{T+1} - \boldsymbol{\alpha})^\top = \eta_Y \boldsymbol{f}_n (\boldsymbol{e}_n - \boldsymbol{\alpha})^\top.$$

Note that for three-token sequences, $\boldsymbol{f}_n$ is a one-hot vector, and thus $\dot{Y}$ has only one non-zero row, and each row of $Y$ can be analyzed independently. For the four-token sequences, we use the same reparameterization strategy as in [14], where we denote $W = [\boldsymbol{w}_1, \boldsymbol{w}_2, \ldots, \boldsymbol{w}_K]^\top \triangleq F^\top Y \in \mathbb{R}^{K \times M}$ with $F = [\boldsymbol{f}_1, \ldots, \boldsymbol{f}_K] \in \mathbb{R}^{M \times K}$.

Note that the parameter $W$ can be viewed as a combination of $Y$ and $Z$ where $Z$ is fixed. The following lemma shows the dynamics of $W$, assuming we are performing gradient updates on $W$ instead of $Y$.

**Lemma 12** (Dynamics of $W$). *Assume we perform gradient updates directly on $W$ instead of $Y$ with learning rate $\eta_Y$ and batch size 1, and assume $M \gg 100$ and $\frac{1}{M^{0.99}} \ll \eta_Y < 1$. Let $t \gtrsim \frac{N \ln M}{\eta_Y}$ and let $W(t)_i$ denote the $i$-th row of $W(t)$ and $W(t)_{ij}$ denote the $(i,j)$-th entry of $W(t)$. Then for training sequence $(x_1, \mathtt{R}_1, \mathtt{R}_2, x_4) \in \mathcal{D}_{train}$ at time $t$, we have*

$$W(t)_{x_4, x_4} \gtrsim \ln\left(M \eta_Y t / N\right), \quad and \quad W(t)_{x_4, x} \lesssim -\ln\left(M \eta_Y t / N\right) / M, \quad \forall x \neq x_4,$$

*and for any test sequence $(x_1, \mathtt{R}_1, \mathtt{R}_2, x_4) \in \mathcal{D}_{test}$, we have $W(t)_{x_4, x} = 0, \forall x \in [M]$.*

*Proof.* According to Lemma 3 of [14], for training sequence $(x_1, \mathtt{R}_1, \mathtt{R}_2, x_4) \in \mathcal{D}_{train}$ at time $t$, only the $x_4$-th row of $W$ will be updated and the gradient

$$\dot{\boldsymbol{w}}_{x_4}(t) = \eta_Y (\boldsymbol{x}_4 - \boldsymbol{\alpha}_{x_4}(t))$$

where $\boldsymbol{\alpha}(t) = \exp(\boldsymbol{w}_{x_4}(t)) / (\mathbf{1}^\top \exp(\boldsymbol{w}_{x_4}(t)))$. Therefore, the dynamics of $W$ is nearly identical to the dynamics of $Y$ in Lemma 2, and we can use the proof of Lemma 2 to conclude the results. $\square$

With the dynamics of $W$, we can obtain the following result:

**Proposition C.1** (Reversal curse for the four-token sequences). *Assume we perform gradient updates directly on $W$ instead of $Y$ with learning rate $\eta_Y$ and batch size 1, and assume $M \gg 100$ and $\frac{1}{M^{0.99}} \ll \eta_Y < 1$. Let $t \gtrsim \frac{N \ln M}{\eta_Y}$ denote the time step which also satisfies $\ln t \gtrsim \ln(NM/\eta_Y)$. For training sequence $(x_1, \mathtt{R}_1, \mathtt{R}_2, x_4) \in \mathcal{D}_{train}$ at time $t$, we have*

$$p_{\theta(t)}(x_4 | x_1, \mathtt{R}_1, \mathtt{R}_2) \geq 1 - \frac{M-1}{(M \eta_Y t / N)^c} \to 1, \quad as \ t \to \infty$$

*for some constant $c > 0$, and for any test sequence $(x_1, \mathtt{R}_1, \mathtt{R}_2, x_4) \in \mathcal{D}_{test}$ that is not included in the training set $\mathcal{D}_{train}$, we have*

$$p_{\theta(t)}(x_4 | x_1, \mathtt{R}_1, \mathtt{R}_2) \leq 1/M.$$

*Proof.* For any sequence $(x_1, x_2 = \mathtt{R}_1, x_3 = \mathtt{R}_2, x_4)$ where $T = 3$,

$$
\begin{aligned}
p_{\theta(t)}(x | x_1, \mathtt{R}_1, \mathtt{R}_2) &= \frac{\exp\left(\boldsymbol{x}^\top Y(t)^\top \mathrm{LN}(X^\top \boldsymbol{b}_T)\right)}{\sum_{x' \in [M]} \exp\left(\boldsymbol{x'}^\top Y(t)^\top \mathrm{LN}(X^\top \boldsymbol{b}_T)\right)} \\
&= \frac{\exp\left(\boldsymbol{x}^\top Y(t)^\top \boldsymbol{f}_{x_4}\right)}{\sum_{x' \in [M]} \exp\left(\boldsymbol{x'}^\top Y(t)^\top \boldsymbol{f}_{x_4}\right)} \\
&= \frac{\exp\left(\boldsymbol{x}^\top \boldsymbol{w}_{x_4}(t)\right)}{\sum_{x' \in [M]} \exp\left(\boldsymbol{x'}^\top \boldsymbol{w}_{x_4}(t)\right)} \\
&= \frac{\exp\left(W(t)_{x_4, x}\right)}{\sum_{x' \in [M]} \exp\left(W(t)_{x_4, x'}\right)}.
\end{aligned}
$$

The above next token probability formulation is almost identical to Proposition 4.2 after replacing $Y$ with $W$. Combining the dynamics of $W$ as shown in Lemma 12, we can use the proof of Theorem 3 to conclude the result. $\square$

Finally, we discuss the role of $Z$ in the four-token-sequence settings. Note that Proposition C.1 assumes gradient update on $W$ instead of $Y$. While we are not able to perform gradient updates on $W$ directly, it is equivalent to modifying the gradient of $Y$ to be

$$\dot{Y} = \eta_Y(\boldsymbol{f}_n - FE'\boldsymbol{e}_n)(\boldsymbol{e}_n - \boldsymbol{\alpha}_n)^\top$$

according to Lemma 3 of [14], where the next token $x_{T+1} = n$, $E' = (I+E)^{-1} - I$, $E = F^\top F - I$, and $F = [\boldsymbol{f}_1, \ldots, \boldsymbol{f}_N] \in \mathbb{R}^{M \times N}$ which only contains the next token that appears in the training set. Compared to the original gradient of $Y$

$$\dot{Y} = \eta_Y \boldsymbol{f}_n(\boldsymbol{e}_n - \boldsymbol{\alpha}_n)^\top,$$

one can obtain that the modification of the gradient of $Y$ is small if $\lambda_1(E)$ is small.

Note that $E_{ii} = \|\boldsymbol{f}_i\|_2^2 - 1 = 0$ and $E_{ij} = \boldsymbol{f}_i^\top \boldsymbol{f}_j$. Also, for any sequence $(x_1 = n', x_2 = \texttt{R}_1, x_3 = \texttt{R}_2, x_4 = n)$ in training set,

$$\boldsymbol{f}_n = \text{LN}(X^\top \boldsymbol{b}_T) = \frac{b_{13}\boldsymbol{e}_{n'} + b_{23}\boldsymbol{e}_{\texttt{R}_1}}{\sqrt{b_{13}^2 + b_{23}^2}},$$

where

$$b_{13} = \exp(Z_{\texttt{R}_2,n'}), b_{13} = \exp(Z_{\texttt{R}_2,\texttt{R}_1}).$$

Note that $\texttt{R}_1$ is a common token that appears in each training sentence, and $n'$ is a distinct token that only appears in one training sentence as a contextual token. By Theorem 2 of [14], under certain technical assumptions, $\dot{Z}_{\texttt{R}_2,n'} > 0$ and $\dot{Z}_{\texttt{R}_1,n} < 0$ and one can expect $Z_{\texttt{R}_2,n'} - Z_{\texttt{R}_1,n}$ to be sufficiently large after sufficient time of training. Therefore,

$$\boldsymbol{f}_n = \tilde{b}_{13}\boldsymbol{e}_{n'} + \tilde{b}_{23}\boldsymbol{e}_{\texttt{R}_1}$$

with $\tilde{b}_{23}$ close to 0. Consider a simple case where for each $n \in [N]$, $\boldsymbol{f}_n = \sqrt{1-c^2}\boldsymbol{e}_{n'} + c\boldsymbol{e}_{\texttt{R}_1}$ for $c$ sufficiently small. Then

$$E_{ij} = \boldsymbol{f}_i^\top \boldsymbol{f}_j = (\sqrt{1-c^2}\boldsymbol{e}_{i'} + c\boldsymbol{e}_{\texttt{R}_1})^\top(\sqrt{1-c^2}\boldsymbol{e}_{j'} + c\boldsymbol{e}_{\texttt{R}_1}) = c^2.$$

Therefore, one can calculate that $\lambda_1(E) = c^2(N-1)$. When $c \ll \frac{1}{\sqrt{N}}$, we have $\lambda_1(E) \ll 1$, and thus the gradient update of $W$ and gradient update of $Y$ are almost the same.

## D Experiments for chain-of-thought

In this section, we conduct experiments for COT on multi-layer transformers to validate theoretical results in Section 4.2.

**Dataset construction.** Similar to Section 5, we randomly sample three disjoint sets of entities $\mathcal{A}, \mathcal{B}, \mathcal{C} \subset \mathcal{V}$, and reverse two additional tokens for $\rightarrow$ and $\rightsquigarrow$, respectively. Next, we specify a bijection from $\mathcal{A}$ to $\mathcal{B}$, and a bijection from $\mathcal{B}$ to $\mathcal{C}$ randomly. For each $\texttt{A}_i \in \mathcal{A}$ and its corresponding $\texttt{B}_i \in \mathcal{B}$ and $\texttt{C}_i \in \mathcal{C}$, we can obtain a triple of sequences $(\texttt{A}_i \rightarrow \texttt{B}_i, \texttt{B}_i \rightarrow \texttt{C}_i, \texttt{A}_i \rightsquigarrow \texttt{C}_i)$, and split the set of all triples into training triples and validation triples. All three sequences of a training triple will be added to the training set, while for a validation triple, we add $\texttt{A}_i \rightarrow \texttt{B}_i$ and $\texttt{B}_i \rightarrow \texttt{C}_i$ to the training set and add $\texttt{A}_i \rightsquigarrow \texttt{C}_i$ to the validation set. Therefore, the model will learn both direct and indirect implications for the training triples and only learn the direct implications for each validation triple while being tested on the indirect implication.

**Results.** Figure 3 shows the experiment results for COT using the same model architecture and configurations as in Figure 1 (the training set size is 540, and the validation set size is 60 resulting from 140 training triples and 60 validation triples), which is consistent with Theorem 7. One can refer to Appendix E.3 for additional experiments with various model configurations and vocabulary sizes. We also empirically validate the intransitivity of model weights (i.e., logits) for multi-layer transformers in Figure 4, which shows that for a validation triple $(\texttt{A}_i, \texttt{B}_i, \texttt{C}_i)$ of which only the direct implication "$\texttt{A}_i \rightarrow \texttt{B}_i$" and "$\texttt{B}_i \rightarrow \texttt{C}_i$" appears in the training set, although the weights from $\texttt{A}_i$ to $\texttt{B}_i$ and from $\texttt{B}_i$ to $\texttt{C}_i$ are trained large as indicated by the diagonals of the first two bottom matrices, the weights from $\texttt{A}_i$ to $\texttt{C}_i$ gets hardly trained as indicated by the diagonals of the last matrix. We also emphasize that another reason that COT is necessary is that all tokens $\texttt{A}_i$, $\texttt{B}_i$, and $\texttt{C}_i$ are different tokens with randomly initialized embedding and thus irrelevant. When these tokens are relevant and show specific patterns, the validation loss can also get better. See more details in Appendix E.4.

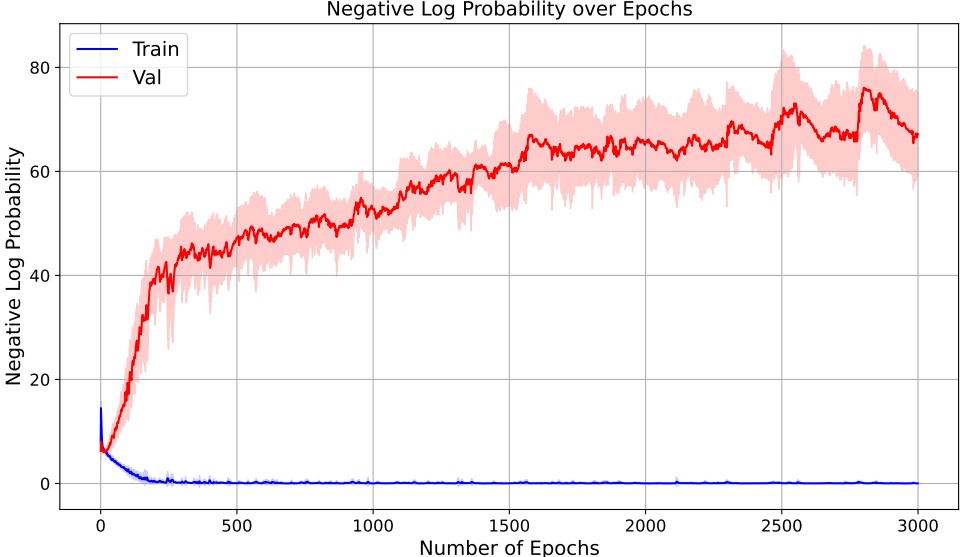

Figure 3: Experiment results of COT under default configuration (see Table 3). The curves represent the (average) negative log probability of the model predicting the next token to be: (1) $B_i$ given the input "$A_i \rightarrow$", (2) $C_i$ given the input "$B_i \rightarrow$", or (3) $C_i$ given the input "$A_i \rightsquigarrow$". Similar to the reversal curse experiment, while the sentences in the training set can be learned nearly perfectly, the model is not able to predict the correct next token in the validation set better than a uniformly random guess. Both curves are averaged over 10 random seeds.

# E  Additional Experimental Results

In this section, we show additional experimental results for Section 5.

## E.1  Details of model architectures and hyperparameters

For both the reversal curse and COT experiments, we used the GPT2 model architecture [63][6] and trained the model with the AdamW optimizer for 3000 epochs of batch size 64. See Table 2 for a full list of hyperparameters. We also conducted experiments under various model configurations and vocabulary sizes to show that the results in Section 5 and appendix D are consistent under different settings. See Table 3 for a complete list of different configurations, where the default choices are boldened. For each curve in all figures, the results are averaged over 10 trials, and the error bar is calculated using standard deviation. We run each trial on an Nvidia A100 GPU and it typically takes 0.5-1.5 hours for each trial.

| Parameters | Values |
|---|---|
| Learning Rate | 0.01 |
| Weight Decay $\lambda$ | 0.9 |
| $(\beta_1, \beta_2)$ | (0.9, 0.999) |
| Batch Size | 64 |
| Number of Epochs | 3000 |

Table 2: Full list of hyperparameters for AdamW optimizer and training.

## E.2  Additional experimental results for the reversal curse

In this section, we show additional experimental results for the reversal curse under different configurations, including different vocabulary sizes (Figure 5), different number of layers (Figure 6),

---

[6]Apache License 2.0

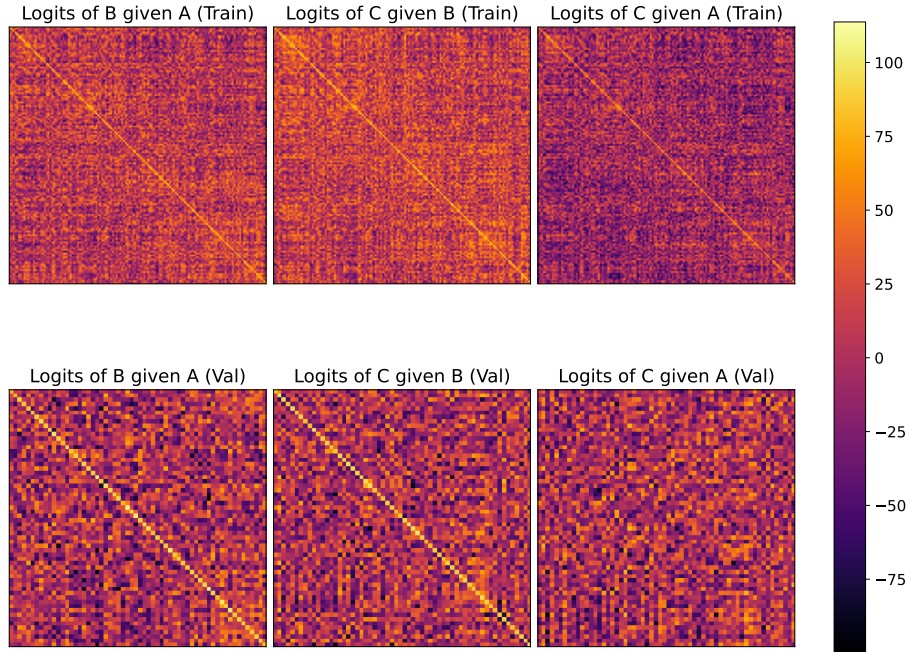

Figure 4: Visualization of the weights (logits) of the model with default configurations trained after 3000 epochs for COT experiment. The matrices are similar to Figure 2. The row tokens for the top matrices are $A_i$, $B_i$, $A_i$ and column tokens are $B_i$, $C_i$, $C_i$ for training triples respectively. Similarly, the bottom matrices correspond to validation triples. For validation triples $(A_i, B_i, C_i)$, the weights from $A_i$ to $C_i$ get hardly trained as indicated by the diagonals of the last matrix.

| Parameters | Values |
|---|---|
| Number of Layers | 12, **24**, 48 |
| Number of Heads | **12** |
| Vocabulary Size | 20, 50, 200, **800**, 2000 |
| Entity Length | **1**, 2, 3 |
| Positional Encoding Type | None, **Absolute**, Relative |
| Token, Positional Embedding | **Learnable**, Frozen |

Table 3: The list of different configurations for experiments in Appendices E.2 and E.3. Default choices are boldened for each row.

different positional encoding (Figure 7), different entity lengths (Figure 8) and whether token and positional embeddings are trainable or fixed (Figure 9). Our experimental results consistently show that the reversal curse happens under different settings.

We provide additional experimental results that (1) the reversal curse still happens even if the embedding dimension is much smaller than the vocabulary size in Appendix E.2.1; (2) the embeddings of different tokens are nearly orthogonal; (3) the reversal curse does not happen under the in-context learning settings.

### E.2.1 The reversal curse under small embedding dimensions

Although for theoretical analysis in Section 3, we assumed the embedding dimension is polynomial in the vocabulary size, in practice, the embedding dimension only needs to be the order of logarithm of the vocabulary size. Figure 10 shows that for a much smaller embedding size, the reversal curse still happens.

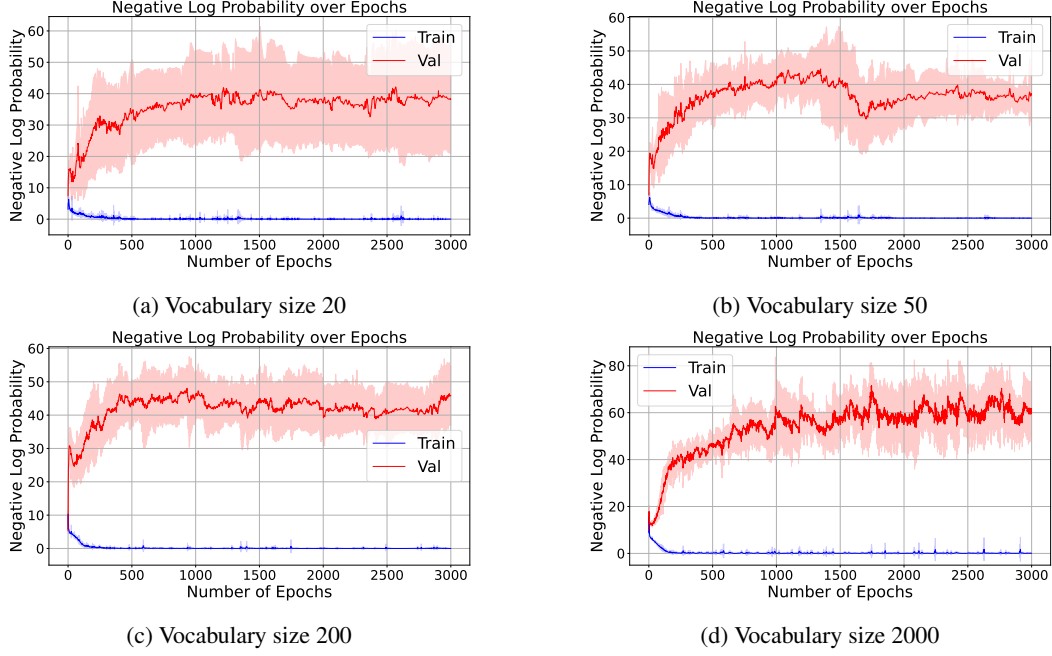

(a) Vocabulary size 20

(b) Vocabulary size 50

(c) Vocabulary size 200

(d) Vocabulary size 2000

Figure 5: Results for reversal curse for different vocabulary sizes. All other configurations are set as default values as in Table 3. The training set sizes for the above four experiments are 9, 20, 85, 850 respectively, and the validation set sizes are 1, 4, 15, 150 respectively.

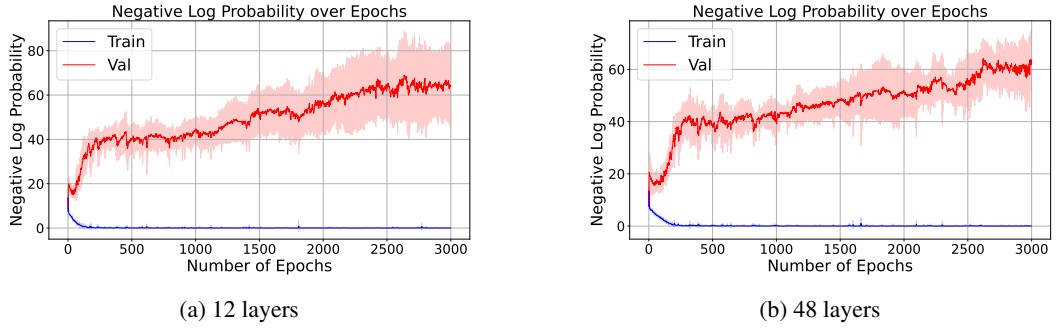

(a) 12 layers

(b) 48 layers

Figure 6: Results for reversal curse for different numbers of layers of the transformer. All other configurations are set as default values as in Table 3.

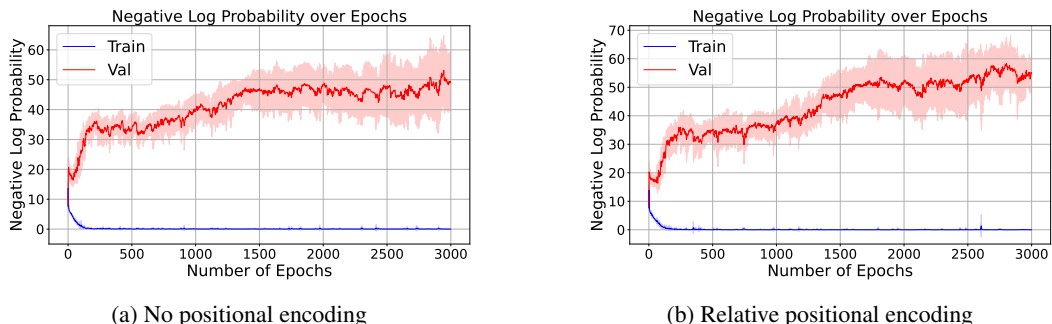

(a) No positional encoding

(b) Relative positional encoding

Figure 7: Results for reversal curse with no positional encoding or relative positional encoding. For relative positional encoding, we follow the Rotary Position Embedding (RoPE) method proposed by [65]. We use the implementation of this repo, MIT license. All other configurations are set as default values as in Table 3.

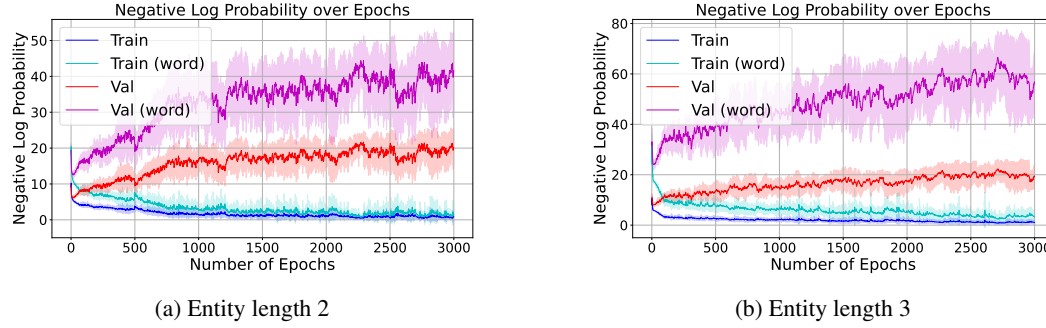

(a) Entity length 2                    (b) Entity length 3

Figure 8: Results for reversal curse with different entity lengths. Each entity $A_i$ or $B_i$ consists of multiple tokens and different entities may have overlapped tokens. The "Train" curve represents the negative log probability of predicting the first token of the output entity, and "Train (word)" represents the negative log probability of predicting all tokens one by one of the output entity. All other configurations are set as default values as in Table 3. The training set sizes for the above two experiments are 680 and 250, respectively, and the validation set sizes are 120 and 50, respectively.

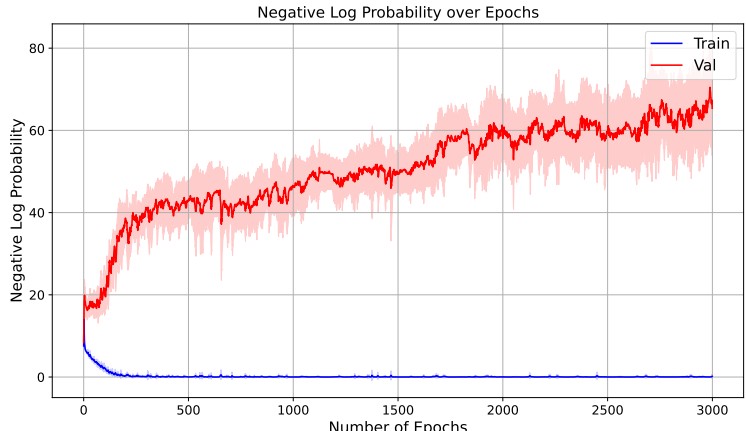

Figure 9: Results for reversal curse with fixed token embedding and fixed positional embedding. All other configurations are set as default values as in Table 3.

### E.2.2    Near orthogonal embeddings

Note that in Section 3, our analysis relies on the fact that embeddings of different tokens are nearly orthogonal, and in Section 4, the embeddings are effectively one-hot. In Figure 11, We show that in practice, even if the embedding dimension is much smaller than the vocabulary size, the near orthogonality condition still holds.

### E.2.3    The reversal curse does not happen in ICL settings

We also emphasize that the reversal curse does not happen in ICL settings, which means if "$A \rightarrow B$" is provided as part of the prompt, then the model is able to answer "$B \leftarrow A$". Figure 12 shows preliminary results of ICL. All the sentences in the dataset have the format of "$A_i R B_j \Leftrightarrow B_j R^{-1} A_i$", which is a seven-token sentence where R and $R^{-1}$ is a pair of relationships inverse to each other, and "$\Leftrightarrow$" is another reserved token representing equivalence. There are ten different $B_j$ and $n$ different $A_i$ where $n = 100$ for the left figure in Figure 12 and $n = 200$ for the right figure. For each $A_i$, we construct ten sentences using different $B_j$ , and we randomly chose three of them to be included in the validation set and seven other sentences in the training set. The result of Figure 12 shows that the reversal curse does not happen during the ICL setting.

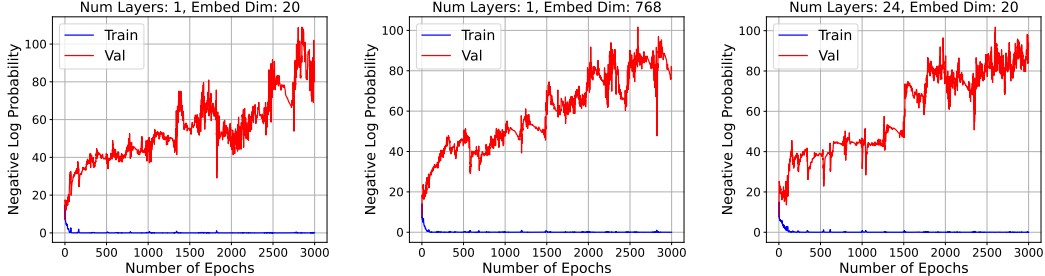

Figure 10: Results for reversal curse for different numbers of layers and different embedding dimensions. All other configurations are set as default values as in Table 3.

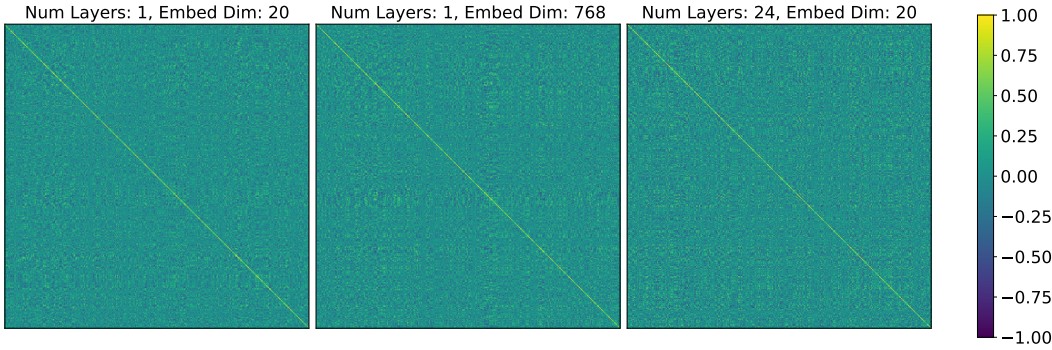

Figure 11: A heat map of cosine similarity between token embeddings (all $A_i$ and $B_i$) after 3000 epochs training under different settings. The settings are the same as Figure 10. Under different numbers of layers or embedding dimensions, most of the non-diagonal entries are close to 0, which shows that the embeddings of different tokens are nearly orthogonal.

### E.3 Additional experimental results for chain-of-thought

In this section, we show additional experimental results for COT under different configurations, including different vocabulary sizes (Figure 13), different number of layers (Figure 14), different positional encoding (Figure 15), different entity lengths (Figure 16) and whether token and positional embeddings are trainable or fixed (Figure 17). Note that our experimental results consistently show the necessity of COT under different settings.

### E.4 Chain-of-thought with relevant tokens

In Appendix D, we briefly mentioned that the irrelevance of different entity tokens is one of the reasons that COT is necessary. Now, we show that if the entity tokens are correlated and show specific patterns, it is possible for a model to deduce indirect implications automatically.

Instead of using single tokens $A_i$, $B_i$, $C_i$ to represent each entity, now we use two tokens $Ai$, $Bi$, $Ci$ to represent entities, where $A$, $B$ and $C$ are three common tokens shared by each triple, and token $i$ are distinct for each triple. Figure 18 shows that for the above version of COT where tokens in the same chain are correlated, the model is able to "deduce" $Ai \rightsquigarrow Ci$ after training on $Ai \rightarrow Bi$, $Bi \rightarrow Ci$ and other training samples.

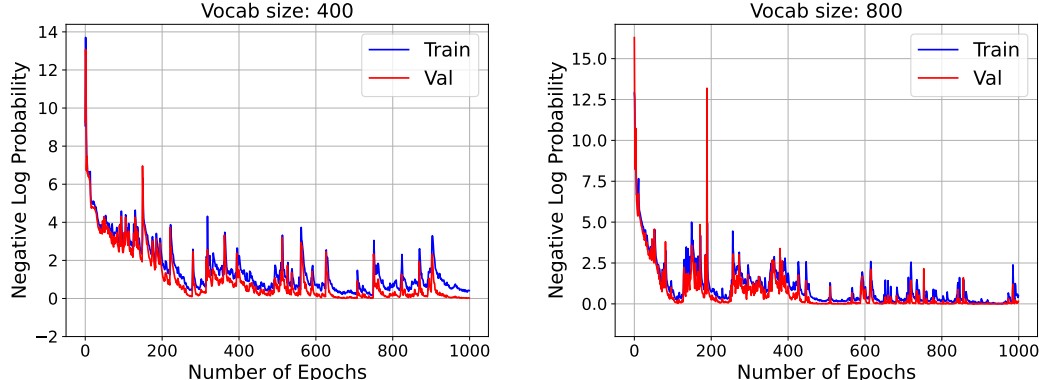

Figure 12: Training and validation loss under in-context learning (ICL) settings. All sentences consist of seven tokens and have the form of "$A_iRB_j \Leftrightarrow B_jR^{-1}A_i$". The loss is calculated on the last token. For the left figure, the training set size is 700, the validation set size is 300, and the vocabulary size is 400; for the right figure, the training set size is 1400, the validation set size is 600, and the vocabulary size is 800. All other configurations are set as default values as in Table 3. The result shows that the reversal curse does not happen in ICL settings, i.e., if "$A_iRB_j$" is provided as part of the prompt, then the model is able to recognize "$B_jR^{-1}A_i$".

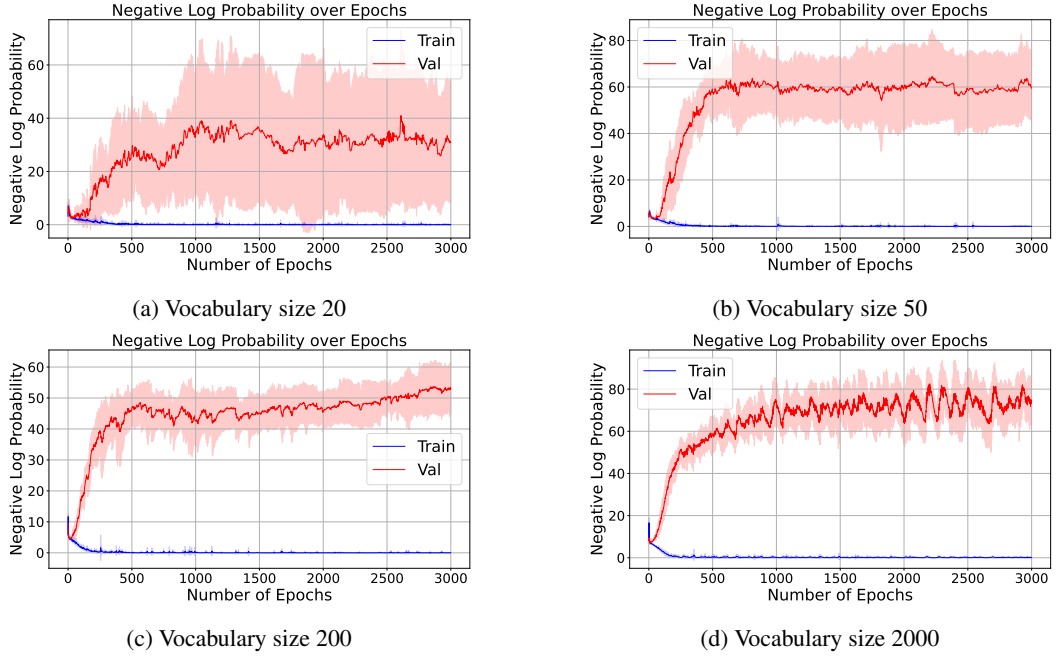

(a) Vocabulary size 20

(b) Vocabulary size 50

(c) Vocabulary size 200

(d) Vocabulary size 2000

Figure 13: Results for COT for different vocabulary sizes. All other configurations are set as default values as in Table 3. The training set sizes for the above four experiments are 14, 32, 135, 1350 respectively, and the validation set sizes are 1, 4, 15, 150 respectively.

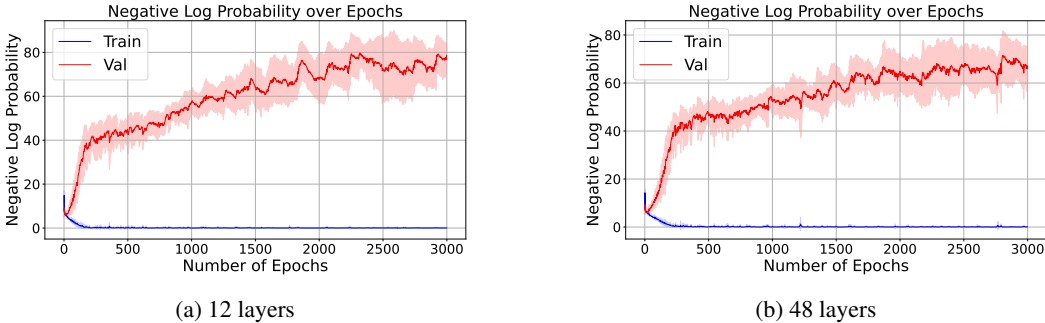

(a) 12 layers  (b) 48 layers

Figure 14: Results for COT for different number of layers of the transformer. All other configurations are set as default values as in Table 3.

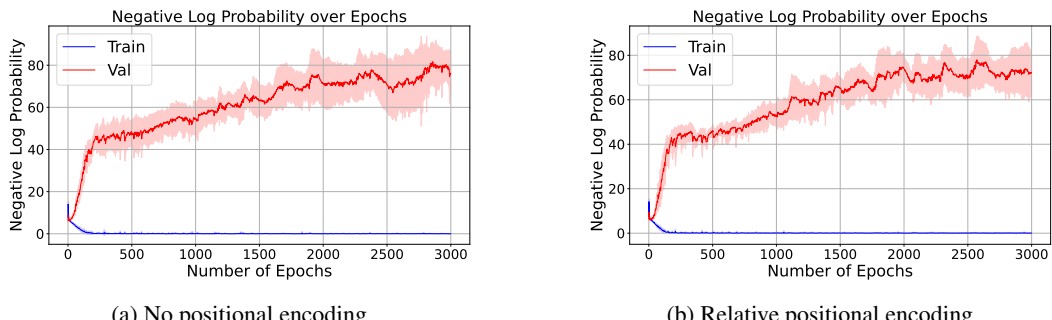

(a) No positional encoding  (b) Relative positional encoding

Figure 15: Results for COT with no positional encoding or relative positional encoding. For relative positional encoding, we follow the Rotary Position Embedding (RoPE) method proposed by [65]. All other configurations are set as default values as in Table 3.

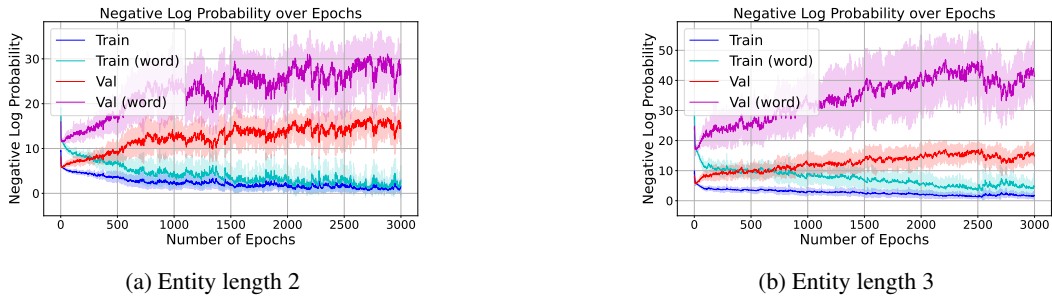

(a) Entity length 2  (b) Entity length 3

Figure 16: Results for COT with different entity lengths. The setting and curves are similar to Figure 8. All other configurations are set as default values as in Table 3. The training set sizes for the above two experiments are 1080 and 400, respectively, and the validation set sizes are 120 and 50.

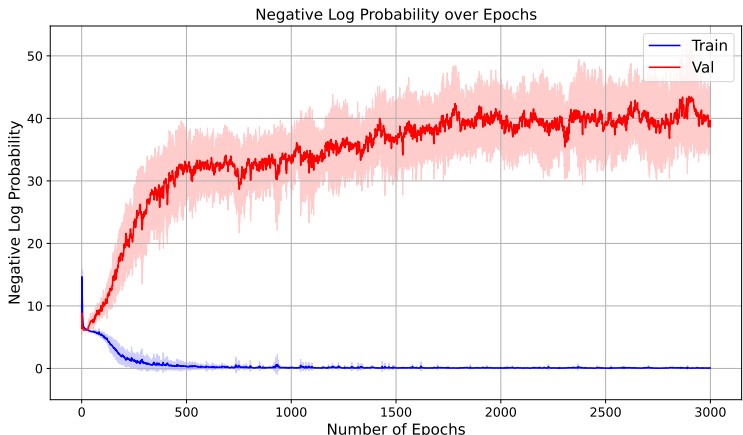

Figure 17: Results for COT with fixed token embedding and fixed positional embedding. All other configurations are set as default values as in Table 3.

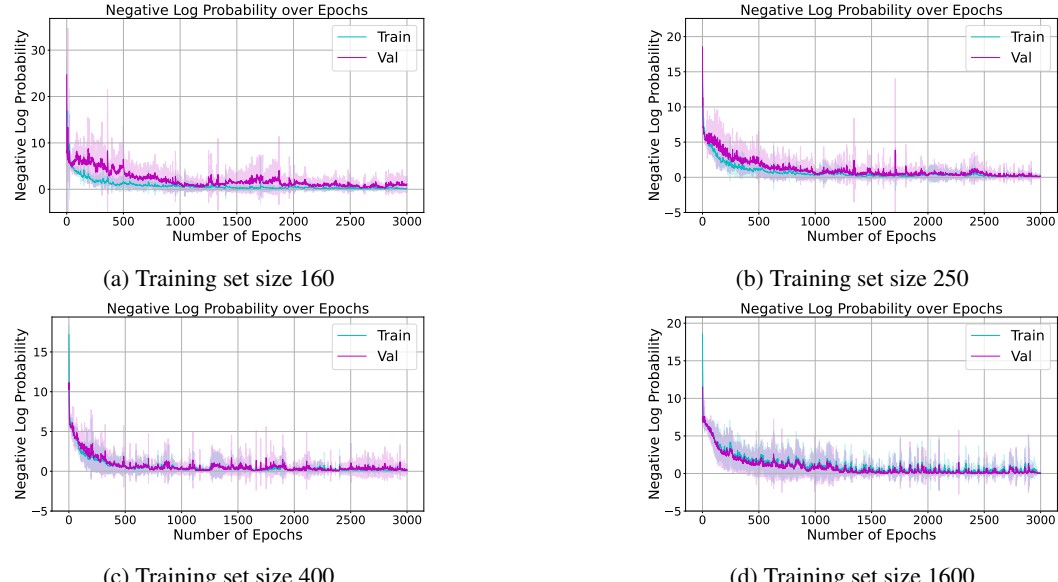

(a) Training set size 160

(b) Training set size 250

(c) Training set size 400

(d) Training set size 1600

Figure 18: Results for COT where each entity is represented by two tokens, i.e., `Ai`, `Bi`, or `Ci`. The validation set sizes are 50. The model is able to "deduce" unseen `Ai` $\rightsquigarrow$ `Ci` by learning underlying patterns.

