# OpenReview forum: "Towards a Theoretical Understanding of the 'Reversal Curse' via Training Dynamics"
_NeurIPS.cc/2024/Conference — NeurIPS 2024 poster_

### Official Review · Reviewer_82qZ · 2024-06-17

**Soundness:** 4
**Presentation:** 3
**Contribution:** 2
**Rating:** 5
**Confidence:** 4

**Summary:**

The paper provides a theoretical analysis of the recently discovered reversal curse phenomenon of LLMs [1], whereby a model trained or fine-tuned on the sentence "A is B" often fails to generalize to the reverse direction "B is A". The authors study the gradient dynamics of two auto-regressive models, a bilinear model and a one-layer transformer, when optimizing for cross-entropy loss of next-token prediction.

The models are trained on sentences containing both implications $A\to B$, $B\leftarrow A$ for train tokens $A,B$ but only one implication $A'\to B'$ for test tokens $A',B'$, and tested on the reverse implication $B'\leftarrow A'$. The paper reveals a strict separation between training and test error, showing that train error can become arbitrarily small while test error is still close to random guessing. The analysis exploits the asymmetry of the reparametrized model weights, where the weights from $A$ to $B$ do not directly affect the weights from $B$ to $A$. An extension to analyzing chain-of-thought and experiments on multi-layer transformers are also given.

[1] Berglund et al. The Reversal Curse: LLMs Trained on "A is B" Fail to Learn "B is A". ICLR 2024.

**Strengths:**

The paper provides a timely study on a fundamental limitation of the reasoning capabilities of LLMs. The analysis is able to incorporate the next-token prediction mechanism with CE loss with a detailed breakdown of the gradient dynamics and sheds light on various aspects of the reversal curse empirically demonstrated in [1]. Moreover, the modification in Section 4.2 for CoT is very interesting and provides a new approach to the theory of CoT, which so far has been mostly limited to complexity-based expressivity analyses.

**Weaknesses:**

* **Meta-learning:** I am concerned that the paper's approach precludes discussion of the meta-learning capability of Transformers. The surprising aspect of the reversal curse is not just that the LLM fails to learn the reverse implication for a single test instance $A'\to B'$, but fails to *generalize* from a large training set containing both $A_i\to B_i$ and $B_i\leftarrow A_i$. While the problem setup follows this setting, the proof depends on the near-orthonormality of the embeddings, which in turn requires the dimensionality to be much higher than the number of relevant tokens. This assumption feels very strong: in effect, it forces the model to disassociate different tokens and reduces learning relationships to encoding specific parameters, e.g. $\Theta_{i,j}$ similar to memorizing a lookup table. The authors then rely on the asymmetry or intransitivity of these 'lookup' parameters to draw their conclusions. However, I think this reduction may prevent a meaningful analysis of generalization behavior in the first place, and does not explain why the reversal curse happens in practical LLMs (where the training dataset is nearly infinite and the model has had ample time to generalize from a continuum of examples).
* **Asymmetry:** Aside from the meta-learning aspect and just looking at the asymmetric encoding perspective, the idea that LLMs store factual associations as directed representations has been empirically proposed multiple times before, see e.g. the references in Sections 3-4 of [1]. The presented analysis, while theoretically very solid and well-detailed, feels to me more like a verification of this 'obvious explanation' in a toy setup rather than providing new insights into the phenomenon, and hence may be less impactful. The authors should include more in-depth discussion on previous empirical studies (the Reversal Curse paragraph in Section 1.1) to better contextualize their contributions rather than just pointing out the lack of theoretical works.
* **In-context learning:** With the above points in mind, it would be interesting to extend the analysis to e.g. theoretically compare the settings of pre-training (or fine-tuning) and in-context learning, given the different training approaches in [1,2] and the observation that ICL manages to overcome the reversal curse [1].

[2] Grosse et al. Studying Large Language Model Generalization with Influence Functions. 2023.

**Questions:**

See weaknesses.

**Limitations:**

Some directions for future work are suggested in the last section. See also weaknesses.

---

> ### Author Rebuttal · Authors · 2024-08-07
>
> We thank the reviewer for their valuable comments. Below are our responses.
>
> **Meta-learning:**
>
> First, we emphasize that the near-orthonormality of the embeddings does not require the dimensionality to be much higher than the number of relevant tokens. In fact, it only requires the dimension $d$ and vocabulary size $m$ satisfies $d \geq \Omega(\log m)$, which means the dimension $d$ can be even much smaller than the number of tokens. This is also empirically verified in our additional experiments (Figure 1 in the global response pdf) The assumption of large dimensionality is only made to show the provable reversal curse phenomenon in bilinear settings (Theorem 2): in large dimensionality, the outer products of the embedding form a $\ell_1$ subspace embedding, which is critical to the theoretical result.
>
> Also, our analysis reveals that given a large training set containing both Ai->Bi and Bi<-Ai where Ai and Bi can be chosen arbitrarily, the model simply learns (or memorizes) these facts/sentences instead of understanding the relationship between ‘->’ and ‘<-’. Therefore, even if the training set contains nearly infinite samples, the model might end up learning an (artificial) mapping between A and B but still fails to generalize to a new pair (Aj, Bj) since Aj and Bj can be arbitrarily chosen.
>
> **Asymmetry:**
>
> We argue that our result is not simply a verification of the obvious explanation but actually provides important new insights. Note that the previous empirical observation that  ‘LLMs store factual associations as directed representations’ is a qualitative and vague argument. However, our analysis quantifies the phenomenon in a rigorous manner: the directed relationship between token $i$ and token $j$ is associated with the logits $Y_{ij}$. Our insights not only explained the reversal curse in a token-level granularity but can also be applied to explain other phenomena such as chain-of-thought. Also, as Reviewer yXHq mentioned, ‘The investigation into the problem, specifically highlighting that the issue stems from training dynamics rather than expressiveness, is particularly insightful. This perspective is refreshing and challenges conventional approaches, potentially inspiring a new stream of research in this direction.’ We also thank the reviewer for the suggestion of including more discussion on previous empirical studies and we will better contextualize our contributions in the revision.
>
> **In-context learning:**
>
> We totally agree that comparing pre-training/fine-tuning and ICL would be an interesting future direction. As suggested also by the Reviewer bwDn, we added additional experiments on ICL (see the ‘In-context learning’ paragraph in the global response).

---

> > ### Comment · Reviewer_82qZ · 2024-08-09
> >
> > Thank you for the detailed reply. I will maintain my score of the submission.

---

> > > ### Author Response · Authors · 2024-08-09
> > >
> > > Thank you for reading our response. We are happy to make further discussion if you have any remaining questions.

---

### Official Review · Reviewer_yXHq · 2024-06-24

**Soundness:** 4
**Presentation:** 3
**Contribution:** 3
**Rating:** 7
**Confidence:** 4

**Summary:**

This paper explores the difficulty auto-regressive LLMs face with simple logical reasoning tasks, such as reversal curse. Through theoretical analysis and experiments, the study examines the training dynamics of gradient descent in bilinear models and one-layer transformers, suggesting that the asymmetry in model weights during training—where learning the relationship from A to B does not ensure learning from B to A—is a significant factor behind the reversal curse. This issue arises from the optimization process under the commonly used cross-entropy loss, which does not enforce symmetry in learning these relationships. The paper also discusses the implications for other logical reasoning tasks, like chain-of-thought reasoning, and validates its theoretical findings with experiments on multi-layer transformers.

**Strengths:**

1. The exploration of the "reversal curse" is a timely and relevant topic within the field. This paper stands out as a pioneering theoretical analysis on this subject, which is commendable.

2. From a technical standpoint, this paper is well-founded. The depth of the analysis is impressive and may hold value not only within the context of this study but also as a reference point for future research. The methodology and the thoroughness of the analysis contribute significantly to the paper's technical soundness.

3. The investigation into the problem, specifically highlighting that the issue stems from training dynamics rather than expressiveness, is particularly insightful. This perspective is refreshing and challenges conventional approaches, potentially inspiring a new stream of research in this direction.

**Weaknesses:**

1. Concerning the problem formulation, I found myself questioning its identifiability (Question 1).

2. Regarding the assumptions made and the proofs provided, I have a few questions that arose during my review (Question 2, 3, 4).

**Questions:**

1. In the context of this paper's system model, two rules merit consideration:
- RULE 1: If A leads to B (A->B), then it is also true that B leads to A (B->A).
- RULE 2: If A leads to B (A->B) and A belongs to a specific set $\mathcal{A}$, then B leads to A (B->A).

The LLMs learn RULE 2. Given that RULE 1 and RULE 2 are indistinguishable under the training dataset, learning RULE 2 instead of RULE 1 is acceptable and not a reverse curse. For example, considering -> as the relationship "mother" and <- as the relationship "son", A->B implies B->A under the condition that B is male.

2. This paper assumes that tokens follow a Gaussian distribution and that only one token is used as the input for attention. I understand that this is used for landscape and generalization analysis. However, these conditions diverge significantly from practical scenarios. Could the authors discuss the applicability of the results in settings where the distribution of tokens is more realistic, and sequences contain more than one token?

3. The explanation for the reversal curse provided between lines 173-179, which employs one-hot vectors for x and y, seems more aligned with current LLM inputs.
- For Theorem 1 and 2, could the proofs be simplified while still achieving similar results if tokens are selected from one-hot vectors?
- Is it possible to apply this intuition to a one-layer Transformer model as well?

4. The paper operates under the assumption that $T$ is limited to 2, and the model considers merely a single input token. This premise leads to a scenario where the attention mechanism is simplified, rendering the problem convex. Does this approach risk oversimplifying the complexities inherent in the model's dynamics? Moreover, is it feasible to extend the analysis to accommodate scenarios involving a greater number of input tokens, such as 2 tokens, to explore how the model behaves under a non-convex setting?

5. Regarding the experimental setup employing a 24-layer 12-head attention Transformer, this configuration seems considerably complex for testing the theoretical findings.
- Could you provide the rationale behind choosing such a large Transformer model?
- I suggest including experiments with a 1-layer Transformer to verify if the observed training dynamics align with the theoretical predictions.
- I am particularly interested in the case where the embedding dimension is (much) smaller than the vocabulary size, specifically for a single-layer Transformer. In this scenario, the Transformer may not have sufficient memory to store all the necessary information. How would this affect the empirical results?

**Limitations:**

Yes

---

> ### Author Rebuttal · Authors · 2024-08-07
>
> We thank the reviewer for their valuable comments. Below are our responses.
>
> **Answer to Question 1**:
>
> Thank the reviewer for pointing out this interesting question. Note that no matter which rule we are choosing, the model is not able to infer B<-A trained on A->B. Although sometimes this is a correct behavior under rule 2, it is not because the model really learns the rule but simply because the model fails to infer B<-A no matter whether A belongs to the specific set. Therefore, the reversal curse happens under both rules. It would be an interesting future direction to anlyze the difference between model behaviors under these two rules.
>
> **Answer to Question 2**:
>
> Actually,	we only assumed the Gaussian distribution of token embeddings for bilinear settings, and the actual condition we used in our proof is that those embeddings are nearly orthogonal to each other, which is empirically validated in our experiments (see Figure 1 in the global response pdf). Therefore, the result is applicable to practical LLM token embedding distributions. For sequences containing more than one contextual token, see responses to Question 4.
>
> **Answer to Question 3**:
>
> Yes. If tokens are one-hot vectors, the proofs can be simplified since we don’t need Lemma 7 which essentially shows that ‘near-orthonormal’ embeddings behave nearly identically to the one-hot embeddings. Also, we don’t need the embedding dimension $d$ to be as large as $poly(m)$. The intuition between lines 173-179 can be applied to (and actually is the same as) a one-layer transformer, where the model parameters $\Theta_{ij}$ in bilinear setting corresponds to the logits $Y_{ij}$ for transformers.
>
> **Answer to Question 4**:
>
> We study the case where $T=2$ in our main results since we aim to more clearly convey the core reason of the reversal curse, i.e., the asymmetry of model weights.  Our analysis can definitely be extended to scenarios involving a greater number of contextual tokens. Actually, we discussed the case $T=3$ in Section 4.3 and provided analysis for $T=3$ in Appendix C.3.
>
> **Answer to Question 5**:
>
> The reason we chose a large transformer model for experiments is that theoretically we proved the reversal curse for one-layer transformers and revealed a core reason for the reversal curse, i.e., model weights asymmetry, so we aim to empirically verify that the result still holds for multi-layer transformers. We added experiments with 1-layer transformers and also the case where the embedding dimension is much smaller than the vocabulary size, and the result still holds (see Figures 1 and 2 in the global response pdf). Actually, the embedding dimension $d$ only needs to be as large as the log of vocabulary size to satisfy the near-orthogonal property. We assumed $d = poly(m)$ in the proof of bilinear setting solely for a rigorous theoretical guarantee, and in practice, $d \gtrsim \log m$ suffices.

---

> ### Comment · Reviewer_yXHq · 2024-08-08
>
> I appreciate that the rebuttal has addressed most of my concerns. However, I am still unclear on how the analysis presented in Appendix Section C.3 extends beyond a simple convex setting, as mentioned in Question 4. Could you please clarify if this would result in an oversimplification? I am inclined to increase my score once this matter is resolved.

---

> > ### Author Response · Authors · 2024-08-08
> >
> > We thank the reviewer for reading our response and the further questions. In Section C.3, we assumed that the learning rate of the decoder layer $Y$ is much faster than the learning rate of the attention layer $Z$, and used another reparameterization $W = F^TY$. For a general setting with contextual token sequence length $T \geq 2$, one should
> > 1. Analyze the dynamics of $W$.
> > 2. Use $W$ to derive the dynamics of $Y$.
> > 3. Use dynamics of $Y$ to further analyze the dynamics of $Z$.
> >
> > The analysis of $Z$ is non-convex and definitely doable, as shown by previous work [1]. However, for the specific question that we are investigating in this paper (i.e., the reversal curse), it suffices to only analyze the dynamics of $Y$ (when $T=1$) or analyze the dynamics of $W$ (when $T \geq 2$). Note that this does not mean the dynamics of $Z$ do not matter. Actually, $W$ is determined by $Y$ and $F$, and $F$ is calculated using $Z$, so the dynamics of $W$ implicitly incorporate dynamics of both $Y$ and $Z$. One can definitely follow the above steps 1-3 to analyze the reversal curse or other settings, where the problem is non-convex, but for this paper, it would be more convenient to directly analyze $Y$ or $W$. Therefore, our method does not risk oversimplifying the complexities of the problem or the model dynamics.
> >
> > **References:**
> >
> > [1] Yuandong Tian, Yiping Wang, Beidi Chen, and Simon Du. Scan and snap: Understanding training dynamics and token composition in 1-layer transformer, 2023

---

> ### Comment · Reviewer_yXHq · 2024-08-09
>
> I have adjusted the score accordingly. I hope this paper will serve as a starting point for the training dynamic analysis in higher-level intelligence tasks, such as reasoning and planning, within LLMs.

---

> > ### Author Response · Authors · 2024-08-09
> >
> > We sincerely thank the reviewer for recognizing our work and raising their score. Also, we thank the reviewer again for their great effort in reviewing our paper, reading our response, and providing helpful and insightful feedback.

---

> > > ### Comment · Reviewer_yXHq · 2024-08-13
> > >
> > > The discussion between you and Reviewer bwDn has reminded me of a question that I had forgotten before. Regarding question 1, could you provide more insight into the relationship between your results and the experiments conducted in [1]? Given that the weights in [1] are asymmetric, I am curious whether the model learns RULE 1, or if it simply increases the size of the specific set $\mathcal{A}$ in RULE 2. Please note that your response to this question will not influence my decision, as I acknowledge that my comment is being posted quite late and I had forgotten this question earlier.
> > >
> > > [1] Reverse Training to Nurse the Reversal Curse

---

> > > > ### Author Response · Authors · 2024-08-13
> > > >
> > > > Our intuition/guess of [1] is that the model did not really learn RULE 1 or RULE 2. For a training sentence A->B, the model learns a large weight from A to B, and by reverse training, the model also learns a large weight from B to A, and therefore, the model is able to infer B<-A. Our paper shows that in the three-token-sentence setting of 3.1 in [1] (which assumed RULE 1 as the underlying ground-truth rule), the above intuition is true and we provided theoretical proof for one-layer transformers. However, it requires more theoretical analysis/experiments to further validate the above intuition/guess in a more general setting.
> > > >
> > > > [1] Reverse Training to Nurse the Reversal Curse

---

### Official Review · Reviewer_bwDn · 2024-07-04

**Soundness:** 2
**Presentation:** 2
**Contribution:** 2
**Rating:** 5
**Confidence:** 4

**Summary:**

This work analyzes the optimization trajectory of (a) a bilinear model trained on sequences of random Gaussian vectors, and (b) a simplified one-layer transformer on sequences of three tokens. It studies how well the trained model generalizes to sequences in reverse order. This study is motivated by an empirical phenomenon observed in large language models, called the "reversal curse." Furthermore, the paper considers some variations of the main simplified settings, notably a setup that mimics the so-called chain of thought technique. Finally, experiments are presented that demonstrate the validity of the theoretical results.

**Strengths:**

This work studies a well-documented empirical phenomenon of autoregressive language models, namely their inability to operate as databases (since the order of a query matters) - a phenomenon called "the reversal curse." A better theoretical understanding of the subject could have a substantial impact. The current work appears to be the first to study it theoretically from an optimization perspective. The paper has certain merits: in particular, it offers an interesting theoretical analysis of training dynamics of simplified models on synthetic data. The proven theorems/propositions seem correct to me (I did not manage to verify all the steps - see questions on how the presentation of some results could be improved), and the techniques, whilst appearing to be standard, are interesting.

**Weaknesses:**

Unfortunately, I identified the following issues with the paper in its current form:

* Unclear motivation of the theoretical framework: As far as I understand, the so-called "reversal curse" is interesting for cases where we expect a model to have deduced the transitivity of a relation before evaluating. For instance, this could have happened from pretraining on some large corpus and then fine-tuning/testing on Question-Answer data, like the experiments considered in [1]. However, the setup in this work only considers fine-tuning on well-structured, synthetic datasets. As a result, a constant question that I had in mind while reading the theoretical part of this paper was "why should I expect the contrary of what is proven?". Is there any expectation from a model to learn a reverse relation solely from training on only one direction? The answer seems obviously to be "no", in the same way that we do not expect the probabilities of other unseen sentences to increase. I would appreciate it if the authors could elaborate on that in their rebuttal and perhaps add a discussion of it in the paper. Ideally, an analysis of pretrained models would have been appropriate for theoretically studying this phenomenon.

* Imprecise writing: The paper is not very well written in places, and it is hard to follow with terms that are not being defined before usage. For example, lines 35-51 try to motivate the paper by arguing that a specific bias in the architecture weights could potentially resolve the "reversal curse," but they quickly dismiss this proposal on the (correct!) basis that this would bias the model in learning nonsense inputs. Thus, I do not see why this argument is worth mentioning. In the same passage, the authors use the term "symmetric loss" without defining it before. It becomes clear to someone what the authors mean after reading the paper and why someone (naively) might consider this as an option, but a priori, it is not clear and does not help with the understanding of the paper. See also questions for specific questions/suggestions on other parts of the paper (and in particular in the technical part of the proofs).

1. The Reversal Curse: LLMs trained on "A is B" fail to learn "B is A". Lukas Berglund, Meg Tong, Max Kaufmann, Mikita Balesni, Asa Cooper Stickland, Tomasz Korbak, Owain Evans.

**Questions:**

Questions/suggestions:

* Define terms used in the text: examples include the term "symmetric loss" in line 47, the term "reparameterization" in line 59.
* Consistent notation: In Section 2, line 130, you define input sequences of tokens $x = (x_1, \ldots, x_T)$, but in the next page you redefine (or specify) input sequences as $(x_i, y_i)$ or $(y_i, x_i)$ (line 144). While it is not a major issue, being consistent would help readers here.
* Can you please clarify the sentence in line 297-298 "Note that ... will not attend to itself". It is not clear to me how the query token does not attend to itself in the adopted framework.
* A criticism of Theorems 1, 2 could be that they rely on the large dimension of the matrix $\Theta$ to basically make it operate as an associative memory (if I understand correctly). Could you please comment on that and perhaps discuss this in a revised version? How does the dimensionality of the model affect the conclusions? I understand that it needs to be large enough, but should we expect the same   conclusions to hold for models of smaller dimensionality?
* The proofs of the Theorems and Propositions appear in a convoluted order in the Appendix. For instance, the proof of Theorem 1 (Theorem 5 in the Appendix) starts almost immediately by invoking Lemma 3 and Lemma 4 (with no commentary), which in turn immediately invokes Lemma 5. Thus, it is difficult to follow and verify the logic of the conclusions. Please consider presenting the proofs with more details and/or change the order of presentation.
* line 625: It is not clear why you set $u = 0.2$.
* line 626: Can you please add some explanation on the first inequality? You probably use the fact that the matrix $\Theta$ is Gaussian, but there exists no justification for that. As far as I understand this the only element of the proof that really makes use of the properties of $\Theta$ so there is value in being explicit about your proof arguments.
* line 641, 'Using Lemma 9': Can you please elaborate on that? The mentioned Lemma shows a result for the norm of a Gaussian vector, not for the inner product of two different vectors.
* Lemma 10 in page 22: A little bit more care should have been taken in the proof of this Lemma. In particular, below line 653, an inequality for the inverse function is being stated. However, there is no guarantee that the function cannot attain the 0 value at some point $t$ (in fact, $f(0) > 0$ and $f'(0) < 0$).
* Theorem 4: I do not understand the conclusion of this theorem. This theorem asserts that a certain conditional probability is no better than random chance, but it does not demonstrate a **neccessity** of chain-of-thought. A statement about the necessity of chain-of-thought should prove that no matter which other inference technique we use, if it is not chain-of-thought then it will not succeed. This is very different from the proved theorem. Can you please address this in your rebuttal?
* Suggestion on experiments: An interesting observation of [1] is that the "reversal curse" is remedied if the reverse sentence appears in context. Do you believe that a similar observation could be made in your synthetic setup? If the answer is affirmative, then this would actually alleviate some of the current concerns with the validity of the framework in explaining the "reversal curse".
* Figure 15 is interesting, the authors could perhaps consider add some discussion of it in the main paper.

A few typos/grammatical errors I caught:

* line 87: "phenomenonin" -> "phenomenon in"
* line 88: "Several paper studies".
* line 98: "There are rich literatures"
* line 117: "We use $poly(x_1, \ldots, x_n)$ to denote the polynomial of $x_1, \ldots, x_n$." -> what polynomial? you mean **a** polynomial function.


1. The Reversal Curse: LLMs trained on "A is B" fail to learn "B is A". Lukas Berglund, Meg Tong, Max Kaufmann, Mikita Balesni, Asa Cooper Stickland, Tomasz Korbak, Owain Evans.

**Limitations:**

Please consider being more elaborate on the NeurIPS paper checklist and your justifications of your answers (the paper currently mostly points to Sections of the paper, without any further justification).

---

> ### Author Rebuttal · Authors · 2024-08-07
>
> We thank the reviewer for their valuable comments. Below are our responses.
>
> > Unclear motivation of the theoretical framework: ...
>
> First, we want to emphasize that according to our analysis in this work, the reversal curse is hard to mitigate even on a well-structured , synthetic dataset both theoretically and empirically, which implies that the reversal curse is even harder to mitigate in a more complex real-world dataset. Therefore, our setup is appropriate for revealing the most essential reason of the reversal curse.
>
> Also, while we definitely agree that we cannot expect a model to learn a reverse relation solely from training in only one direction, we emphasize that our training data contains both directions. Ideally, we expect the model to learn the reverse relationship between ‘$\to$’ and ‘$\gets$’ from training samples ‘$A\_i\to B\_i$’ and ‘$B\_i\gets A\_i$’, and then infer ‘$B_j\gets A_j$’ given ‘$A_j\to B_j$’ for new entity tokens $A_j$ and $B_j$, which is the same as what we would expect for human.
>
> For the analysis of pertained models, in fact, our result for bilinear models also works if the model parameter $\Theta_0$ is initialized to a pre-trained value satisfying the condition $\frac{1}{2m} < p_{\Theta_0}(y_i|x_i), p_{\Theta_0}(x_i|y_i) < \frac{2}{m}$ (as shown in Theorem 5). This implies that the reversal curse would also happen in the pre-trained models. However, we also want to point out that we haven't considered the learning of embedding in our theoretical study, which may lead to further generalization in the pre-trained model (and might be a (partial) solution to the reversal curse).
>
> > Imprecise writing: ...
>
> Thank you for your suggestions and we will incorporate them in the revision.
>
> > Define terms used in the text: examples include the term "symmetric loss" in line 47, the term "reparameterization" in line 59.
>
> One example of the symmetric loss is $L = \sum_{t=1}^T \log p(x_t|x_{1…t-1}) + \log p(x_t|x_{T…t+1})$ and more broadly, it can be defined as $L(x_{1…T}) = L(x_{T...1})$ for any sequence. Reparameterization is a general term, and in this paper, it refers to $Z = UW_QW_K^TU^T/\sqrt{d}$ and $Y = UW_V^TU^T$. We will make these terms more clear and easier to understand in the revision.
>
> > Can you please clarify the sentence in line 297-298 "Note that ... will not attend to itself" ...
>
>  In Line 196, the attention score vectors $b_T$ only contains $b_{1,T}, …, b_{T-1,T}$ which are the attentions of contextual tokens $x_1, … x_{T-1}$ attended by the query token $x_T$. Note that this framework does not take into account $b_{T,T}$, which is the score that $x_{T}$ attends to itself.
>
> > A criticism of Theorems 1, 2 could be that they rely on the large dimension of the matrix $\Theta$...
>
> Please check the 'Small embedding dimension and number of layers' section in the global response.
>
> > The proofs of the Theorems and Propositions appear in a convoluted order in the Appendix ...
>
> Thank you for your valuable suggestions. In revision, we will improve the structure and readability of the proof.
>
> > line 625: It is not clear why you set 𝑢=0.2.
>
> $v$ is set to make sure that $p_\Theta(x_i|x_j)$ is of order $1/n$. The proof works for any small constant.
>
> > line 626: Can you please add some explanation on the first inequality? ...
>
> Fixing $x_i,x_j$, the logit $l_{\Theta}(x_j|x_i)$ is a Gaussian random variable and the bound follows from standard concentration inequality for Gaussians. In revision, we will make this argument clearer.
>
> > line 641, 'Using Lemma 9': Can you please elaborate on that? ...
>
> In line 640, we use the rectangular trick to reduce vector inner products to vector norms. Since the vectors $x_i+x_j$ and $x_i- x_j$ are also Gaussians, we apply Lemma 9 to bound their norms and thus by the right-hand side of line 640, we can bound the inner products.
>
> > Lemma 10 in page 22: A little bit more care should have been taken in the proof of this Lemma ...
>
> Thank you for your valuable suggestion. The functions cannot attain the 0  value because letting $\tau = \inf$  \{ $t: f(t) \leq 0$ \}, a contradiction would happen in the interval $(0,t)$ by the same argument of the proof. We will add this point in the revision.
>
> > Theorem 4: I do not understand the conclusion of this theorem ...
>
> Thanks for pointing this out. We agree that it would be more appropriate to change the ‘necessity of COT’ to the ‘importance of COT’ since Theorem 4 does not exclude all other inference techniques than COT. The main point we want to emphasize here is that the model fails to directly output an indirect conclusion.
>
> > Suggestion on experiments: An interesting observation of [1] is that the "reversal curse" is remedied if the reverse sentence appears in context ...
>
> Thanks for your suggestions on experiments. A similar observation can be made in our settings, and please check the ‘In-context learning’ paragraph in the global response.
>
> > Figure 15 is interesting, the authors could perhaps consider add some discussion of it in the main paper.
>
> Thanks for your suggestions. We will use the additional content page for the camera-ready version to add more discussions.
>
> > A few typos/grammatical errors I caught: ...
>
> Thank you for catching these typos, and we will fix them in the revision.
>
> **References:**
>
> [1] The Reversal Curse: LLMs trained on "A is B" fail to learn "B is A". Lukas Berglund, Meg Tong, Max Kaufmann, Mikita Balesni, Asa Cooper Stickland, Tomasz Korbak, Owain Evans.

---

> > ### Comment · Reviewer_bwDn · 2024-08-13
> >
> > Thank you for your reply, and I apologize for the delay in my response. I appreciate the clarifications, and I hope they are included in the revised paper as well (especially the COT part).
> >
> > > Also, while we definitely agree that we cannot expect a model to learn a reverse relation solely from training in only one direction, we emphasize that our training data contains both directions. Ideally, we expect the model to learn the reverse relationship between ‘$\to$’ and ‘$\gets$’ from training samples ‘$A_i\to B_i$’ and ‘$B_i\gets A_i$’, and then infer ‘$B_j\gets A_j$’ given ‘$A_j\to B_j$’ for new entity tokens $A_j$ and $B_j$, which is the same as what we would expect for human.
> >
> > I disagree with the conclusion. The reverse relationship is not always true. Humans understand that when 'A is B', 'B is A' is not always true (e.g., 'a cat is black', but 'black is not a cat').
> >
> > The motivation of the paper still remains weak/unclear to me, and this is why I am maintaining my score. I appreciate the ICL experiments, and I am sure that they strengthen the paper.

---

> > > ### Author Response · Authors · 2024-08-13
> > >
> > > We thank the reviewer for reading our response, recognizing our clarification and ICL experiments, and giving further comments.
> > >
> > > Regarding the motivation of our work, we agree that there are two possible cases/rules as also mentioned by Reviewer yXHq:
> > >
> > > 1. RULE 1: If A->B, then it is always true that B<-A.
> > >
> > > 2. RULE 2: If A->B, and A/B belongs to specific sets, then B<-A (which means it is not always true that B<-A).
> > >
> > > An example of RULE 1 is ‘A is the parent of B’ is equivalent to ‘B is the child of A’. The example pointed out by the reviewer that 'a cat is black', but 'black is not a cat' is under RULE 2 which requires that B is not an adjective. However, no matter which rule we are choosing, the model is not able to infer B<-A trained on A->B. Although sometimes this is a correct behavior under rule 2 since under rule 2 the reverse is not always true, it is not because the model really learns the rule but simply because the model fails to infer B<-A no matter whether A/B belongs to the specific sets (due to model weights asymmetry). Therefore, the reversal curse happens under both rules, and our paper theoretically explained why the reversal curse happens when we do not expect it to happen (i.e., it happens under RULE 1). It would be an interesting future direction to anlyze the difference between model behaviors under these two rules.
> > >
> > > Please feel free to let us know if you have any remaining questions. We will also make sure to include our clarifications in the rebuttal (especially the COT part) in the revised paper.

---

> > > > ### Comment · Reviewer_bwDn · 2024-08-13
> > > >
> > > > Thank you for your very quick reply. I hope that my feedback and our discussion will help you improve the presentation of your work.

---

> > > > > ### Author Response · Authors · 2024-08-13
> > > > >
> > > > > We are happy that we have addressed your question and thank you again for your helpful feedback. We will polish our work according to your feedback and our discussion.

---

### Official Review · Reviewer_8So4 · 2024-07-12

**Soundness:** 3
**Presentation:** 3
**Contribution:** 3
**Rating:** 7
**Confidence:** 3

**Summary:**

This paper works toward a theoretical explanation of why the "Reversal Curse" occurs in language models. This "curse" refers to the empirical phenomenon that even when trained on certain relationships between entities ("Sue is an aunt of Alice"), the models fail at reversing those relationships in a semantically equivalent way ("Alice is a niece of Sue").
A priori, we might expect that the training dynamics of large language models on large amounts of data would encourage a model that places a high probability on the forward relationship to do the same on the reverse one. But empirical results show this is not the case, and this paper attempts to break down the training dynamics to give a theoretical explanation why.

- Section 3 shows that in a synthetic setting with a bilinear model, the "reversal loss" (predicting B<--A when the training data only contains A-->B for that example) is lower-bounded by a function of the training loss. For certain parameter regimes, this lower bound is 1, indicating that the reversal loss does not decrease due to the training dynamics even when the forward relationship A-->B is present in the training data.

- Section 4 extends this analysis to one-layer transformers, a model also studied in prior work in understanding transformer training dynamics. It also extends the analysis of the reversal curse to show why Chain-of-Thought reasoning is necessary.

**Strengths:**

- Analyzes the training dynamics of increasingly more complex transformer-like models to show why the reversal curse emerges.

- The techniques used in the paper can also be used to analyze why language models sometimes can't distill chain-of-thought reasoning: even though they can directly learn A-->B and B-->C, they can't always directly predict A-->C.

- On a technical level, this paper relaxes some assumptions from other work on training dynamics of the bilinear model in Section 3.

**Weaknesses:**

- If I understand correctly, the ultimate conclusion is that there's no real hope beyond reversal training, i.e., having both A-->B and B-->A in the training data. It would be stronger if the analysis revealed some kind of (partial?) solution to the reversal curse.

**Questions:**

- L42 "Since it is intractable to manually hard-code the constraints to the model parameter, one can alternatively expect the model to learn the higher-level regularity by training samples under unconstrained optimization"

    On my first read, I didn't quite understand what this meant. Is the claim that we might learn B <-- A from A --> B samples even without constraints? Or are the authors trying to say that we could just learn from samples of both A --> B and B <-- A? The former point is unintuitive (how could this be true? Because of some implicit structure in model parameter space / inductive bias?) and the latter point is arguably clear. It's clear from further reading that the authors mean the former, but it was confusing on a first reading. Mentioning "due to training dynamics" would be helpful here.

- What is $\sigma$ in Theorem 2? Defined in Line 156? It's not clear because L157 also says that I could take the parameters to be some pretrained value satisfying certain conditions. Does Theorem 2 only apply to the first case?

- What does Theorem 2 add beyond the interpretation of Theorem 1 in L161--L162?

- In Section 4, why do we need to assume the query token doesn't attend to itself?

- Is there any reason not to just present the reparameterized form starting in L204? This would make the exposition simpler, though I suppose the connection to the transformer might be less clear.

- In Section 4, what happens if the embeddings are clustered in such a way that the set of entities {Bi} and entities {Ai} are disjoint and (say) linearly separable? Intuitively if we already know what kind of entity a new B is, that might help with reversal? As far as I can tell, this is not the case in Section 3, but does that change in S4?

- L87 "phenomenonin"

**Limitations:**

The conclusion includes a fair discussion of the main limitations of the work (it only considers two- or three-token sequences, and the deepest model is considers is a one-layer transformer).

---

> ### Author Rebuttal · Authors · 2024-08-07
>
> We thank the reviewer for their valuable comments. Below are our responses.
>
> >If I understand correctly, the ultimate conclusion is that there's no real hope beyond reversal training, i.e., having both A-->B and B-->A in the training data. It would be stronger if the analysis revealed some kind of (partial?) solution to the reversal curse.
>
> According to the currently widely-used training paradigm (CE loss for next token prediction using SGD/AdamW, etc., without hard-coding constraints on model parameters), the reversal curse might be unavoidable. Therefore, we might need to change the training paradigm. An ultimate solution could be a method that really enforces the model to understand the higher-level regularity that two relationships are inverse to each other, instead of just learning how to predict the next token. A potential partial solution might be learning in a latent space where the model parameter in this latent space shows symmetry to some degree inspired by our model weights asymmetry analysis. All these are interesting and important future directions.
>
> > L42 "Since it is intractable to manually hard-code the constraints to the model parameter, one can alternatively expect the model to learn the higher-level regularity by training samples under unconstrained optimization" On my first read, I didn't quite understand what this meant. Is the claim that we might learn B <-- A from A --> B samples even without constraints? Or are the authors trying to say that we could just learn from samples of both A --> B and B <-- A? The former point is unintuitive (how could this be true? Because of some implicit structure in model parameter space / inductive bias?) and the latter point is arguably clear. It's clear from further reading that the authors mean the former, but it was confusing on a first reading. Mentioning "due to training dynamics" would be helpful here.
>
> The constraints in L42 refer to hard constraints on model parameters during training/optimization (e.g., a concrete constraint on model parameters $\Theta$ can be $\Theta$ is symmetric, or $\| \Theta \|_2 = 1$). Since some easy-to-implement constraints might hurt the model performance and the correct constraints could be rather complicated and hard to enforce, in practice, we often use unconstrained optimization, and thus the model learns the relationship between ‘->’ and ‘<-’ through lots of examples ‘A->B’ and ‘B<-A’. Thanks for pointing this out, and we will make the statement more clear in the revision.
>
> > What is 𝜎 in Theorem 2? Defined in Line 156? It's not clear because L157 also says that I could take the parameters to be some pretrained value satisfying certain conditions. Does Theorem 2 only apply to the first case?
>
> $\sigma$ is the standard deviation of the model parameter $\Theta\_0$. Theorem 5 shows that the reversal curse would happen (with high probability) under either of these two conditions. Therefore, we say that $\Theta\_0$ can also be initialized as a pre-trained value: as long as it satisfies the condition $\frac{1}{2m} < p\_{\Theta\_0}(y\_i|x\_i), p\_{\Theta\_0}(x\_i|y\_i) < \frac{2}{m}$.
>
> > What does Theorem 2 add beyond the interpretation of Theorem 1 in L161--L162?
>
> Theorem 2 is a direct corollary of Theorem 1, stating more explicitly that with high probability the forward loss is low while the reverse loss is high.
>
> > In Section 4, why do we need to assume the query token doesn't attend to itself?
>
> This assumption is mainly for the convenience of theoretical analysis so that the query tokens and contextual tokens can be disjoint, and this assumption is also adopted by previous work both theoretically [1] and empirically [2]. Our experimental results show that when the query token can attend itself, the conclusion still holds. Moreover, even if we assume the query token can attend to itself, previous results [1] indicate that the query token will gradually put more attention weights to distinct tokens (i.e., the entity token Ai or Bi in our settings) and lose attention to common tokens (i.e., the query token itself in our setting) and thus is effectively the same as our assumption.
>
> > Is there any reason not to just present the reparameterized form starting in L204? This would make the exposition simpler, though I suppose the connection to the transformer might be less clear.
>
> Yes, the reason that we start with the actual parameterization and then the reparameterized form is to draw a more clear connection to the transformer.
>
> > In Section 4, what happens if the embeddings are clustered in such a way that the set of entities {Bi} and entities {Ai} are disjoint and (say) linearly separable? Intuitively if we already know what kind of entity a new B is, that might help with reversal? As far as I can tell, this is not the case in Section 3, but does that change in S4?
>
> In section 4, even if $\{A\_i\}$ and $\{B\_i\}$ are disjoint and linearly separable, the reversal curse still happens. The most straightforward example is when all $A\_i$ and $B\_i$ have one-hot embedding, the two sets are disjoint and linearly separable. Our analysis in section 4 then shows the reversal curse still happens. In practice, the embeddings of different tokens are not exactly one-hot but still nearly orthogonal, which is indicated by Figure 1 in the global response pdf. Therefore, even if the embeddings of two sets are linearly separable, knowing what kind of entity a new B is does not help with reversal.
>
> **References:**
>
> [1] Yuandong Tian, Yiping Wang, Beidi Chen, and Simon Du. Scan and snap: Understanding training dynamics and token composition in 1-layer transformer, 2023
>
> [2] Nikita Kitaev, Łukasz Kaiser, and Anselm Levskaya. Reformer: The efficient transformer. ICLR, 2020.

---

### Author Rebuttal · Authors · 2024-08-07

We thank all the reviewers for their great effort in reviewing our paper and providing helpful and insightful feedback. Below we first address some common questions.

**Small embedding dimension and number of layers:** We first acknowledge that the assumption $d \sim poly(m)$ is a bit loose in Theorem 2. Below we explain the differences between theory and practice in our study for embedding dimensions.

1. On the theory side, there are some confusions among related concepts (near orthogonality, large embedding dimension).

    (a) To satisfy ‘near orthogonality’, the embedding dimension d only needs to grow logarithmically w.r.t the number of tokens m, thanks to the Johnson–Lindenstrauss lemma, (i.e., $d \gtrsim \log m$, see Lemma 8 in Appendix).

    (b) Once the embedding is almost orthogonal, we can further obtain that the outer products of the embedding form a $\ell_1$ subspace embedding where the approximation error is bounded by an $\epsilon$-fraction of **$\ell_1$** norm of embeddings (see the first inequality of Lemma 7).

    (c) For a rigorous theoretical analysis, we further need that the above approximation error is bounded by an $\epsilon$-fraction of **$\ell_2$** norm of embeddings (see the second inequality of Lemma 7), which can be obtained by Cauchy-Schwarz inequality but incurs a polynomial dependence of $d$ on $m$.

    (d) A more intuitive explanation: Johnson–Lindenstrauss lemma actually requires that $d \gtrsim \log m/\epsilon^2$. While in practice, it suffices that $\epsilon$ is a small constant, in our theoretical analysis for the bilinear setting in Theorem 2, we need $\epsilon = 1/poly(m)$, which requires $d \gtrsim poly(m)$.

2. On the empirical side, we provided additional experiments showing that, indeed, an embedding dimension $d \gtrsim \log m$ suffices to lead to near orthogonality of embeddings (Figure 1 in the rebuttal pdf) and observe the reversal curse (Figure 2 in the rebuttal pdf). In particular, when $m = 800$, it suffices that $d = 20 \asymp \log m$ as in the first and third subplots in Figures 1 and 2 in the rebuttal pdf.

3. In conclusion, (a) When $d \gtrsim poly(m)$, both theoretical and empirical results imply near-orthogonality of embeddings and the reversal curse phenomenon. (b) When $d$ is between $\log m$ and $poly(m)$, empirical results still imply both conclusions, while our current theoretical analysis for **bilinear models** can only show near-orthogonality of embeddings.

4. [Special case for one-hot embedding] It is worth mentioning that for analysis in Section 4 for one-layer transformers with one-hot embeddings, $d = m$ suffices for both conclusions. It would be an important future direction to develop more advanced techniques to theoretically analyze the case where $d$ is between $\log m$ and $poly(m)$.


We also showed similar empirical results in additional experiments for one-layer transformers, as requested by Reviewer yXHq. Figures 1 and 2 in the attached pdf show that our results still hold for one-layer transformers and for the case where the embedding dimension is much smaller than the vocabulary size.

**In-context learning**: Although an in-depth study of ICL is beyond the scope of this paper, we provided some preliminary results of ICL in the attached pdf. The setup of the ICL experiments is as follows. All the sentences in the dataset have the format of $A\_i \to B\_j \Leftrightarrow B\_j \gets A\_i$, which is a seven-token sentence. There are ten different $B\_j$ and $n$ different $A\_i$ where $n = 80$ for the left figure in Figure 3 and $n=200$ for the right figure. For each $A\_i$, we can construct ten sentences using different $B\_j$, and we randomly choose three of them to be included in the validation set and the other seven sentences to the training set. We plotted the average training and validation losses on the last token during training dynamics. The results imply that under ICL, the reversal curse will not happen, which is consistent with the empirical observation of [1].

**References**:
[1] The Reversal Curse: LLMs trained on "A is B" fail to learn "B is A". Lukas Berglund, Meg Tong, Max Kaufmann, Mikita Balesni, Asa Cooper Stickland, Tomasz Korbak, Owain Evans.

---

### Decision · Program_Chairs · 2024-09-25

**Decision:**

Accept (poster)

**Comment:**

This paper presents a theoretical analysis of the "Reversal Curse" observed in language models. Specifically, this study examines the training dynamics of gradient descent in bilinear models and one-layer transformers, suggesting that the asymmetry in model weights during training is a significant factor behind the reversal curse. This issue is related to the commonly used cross-entropy loss, which does not enforce symmetry in learning process. The analysis is further extended to other logical reasoning tasks, like chain-of-thought reasoning, and the paper validates its theoretical findings with experiments on multi-layer transformers. The presented analysis provides novel insights into the Reversal Curse phenomenon and the paper is well written.